# Multiscale computational evaluation of *Vitex trifolia* phytochemicals as VEGFR2 inhibitors for targeted breast cancer therapy

Arooj Fatima[1], Muhammad Umer Khan [1]*, Saooda Ibrahim[1], Iqra Khurram[1], Qasim Ammar[1], Raima Rehman[2], Alaa S. Alhegaili[3], Raghad S. Alhejaili[4], Nawal Alsubaie[5], Muhammad Ikram Ullah[6]

**1** Institute of Molecular Biology and Biotechnology, The University of Lahore, Lahore, Pakistan, **2** Centre of excellence in Molecular Biology, University of the Punjab, Lahore, Pakistan, **3** Department of Medical Laboratory, College of Applied Medical Sciences, Prince Sattam bin Abdulaziz University, Al-Kharj, Saudi Arabia, **4** Aneis Al-Khair Medical Laboratories, Al-Madinah Al-Munawarah, Saudi Arabia, **5** Department of Pharmacy Practice, College of Pharmacy, Princess Nourah bint Abdulrahman University, Riyadh, Saudi Arabia, **6** Department of Clinical Laboratory Sciences, College of Applied Medical Sciences, Jouf University, Sakaka, Aljouf, Saudi Arabia

* Muhammad.umer4@mlt.uol.edu.pk, umer.khan685@gmail.com

## Abstract

Breast cancer (BC), the second most common cancer, is a genetically heterogeneous disease driven by angiogenesis, cell growth, metastasis, and oxidative stress. VEGFR2, a key angiogenesis regulator, presents a potential target for inhibiting the angiogenic process essential for tumor growth. The present study aimed to investigate the therapeutic potential of phytochemicals of *Vitex trifolia* to inhibit VEGFR2 for BC. The current study employed extensively *in silico*-based computational tools to assess the binding affinity and interactions through docking and validating through simulations, along with evaluating the molecular characteristics of the phytochemicals of *Vitex trifolia*. The docking results revealed that VT-6 (cynaroside) showed the highest docking score (−14.611 kcal/mol), followed by VT-10 (−13.641 kcal/mol) against the VEGFR2 protein. Significant stable interactions were formed by the key interacting residues of the binding pocket (GLU917, ASP1046, LYS868, CYS919, and GLU885), also highlighted through the SIFT analysis. Additionally, the density functional theory (DFT) analysis demonstrated balanced electrophilic and nucleophilic electronic distribution and reactivity for VT-6. The simulations further validated the stability of VT-6 within the binding cavity of VEGFR2, exhibiting flexibility within a range of <3Å and stable conformational changes. Moreover, the MM/GBSA calculations also signify that VT-6 exhibited stronger binding affinity with more negative free energies (−32.5 kcal/mol MM/PBSA, −34.7 kcal/mol MM/GBSA). Notably, principal component analysis (PCA) and free energy landscape (FEL) indicated that the VT-6 complex remained compact during 200 ns simulations. Conclusively, these findings underscore VT-6 as a potent VEGFR2 inhibitor against BC. However, optimizing the

**Data availability statement:** All relevant data are within the paper and its Supporting Information files.

**Funding:** The author(s) received no specific funding for this work.

**Competing interests:** The authors have declared that no competing interests exist.

ADMET profile through structural modifications and nanocarrier delivery, along with *in vivo* and *in vitro* experimental validation, will enhance its therapeutic potential.

## 1. Introduction

Among women, breast cancer (BC) is the most common and second highest cancer as a reason for mortality. The mortality rate increases every year, with more than 250,000 women diagnosed with breast cancer each year in the United States [1]. At the age of 50, there is a lower survival rate than those who had breast cancer at the age of 50–75. The prevalence of BC in women is 100 times higher than in men [2–4]. Multiple factors can be involved in the development of BC. It can be due to internal (genetic) or external (epigenetic) factors like hormonal factors, genetic changes, radiation exposure, family history, obesity, poor lifestyle, reproductive age, late menopause, and use of tobacco use [5].

The progression of BC is characterized by a tumor microenvironment (TME) that helps to grow the cancer cells and acts as a shield for these cells by preventing them from apoptosis. The key factor in the development of BC is angiogenesis (formation of new blood vessels). Angiogenesis is regulated via Vascular endothelial growth factor receptor 2 (VEGFR2), which is secreted by cancer cells. This process nourishes the cancer cells and promotes their proliferation, leading to increased angiogenesis, which is associated with a low survival rate in breast cancer patients [6,7].

Metastasis is a multistep invasion, infiltration, and colonization process of target organs. During the process, the cancer cells undergo phenotypic transformation, most notably epithelial-mesenchymal transition (EMT) to accommodate the evolving microenvironment. These changes are promoted by the TME by providing tumor cell, endothelial cell, and fibroblast interactions to promote proliferation, angiogenesis, and metastasis of breast cancer. Pro-inflammatory factors released by TME can trigger epigenetic alterations in chromatin, thereby promoting tumor initiation and metastasis [8].

For the first time in the 1990s, the receptor of vascular endothelial growth factor (VEGF) was discovered and the receptor was the VEGFR2. A previous study explained that VEGFR2 regulates angiogenesis, which is a process that increases the permeability of vessels [9]. Angiogenesis plays an important in physiological processes such as embryonic development, wound healing, and enhanced vascular permeability [10]. However, VEGFR2 also plays a crucial role in pathological conditions such as cancer, where cancer cells upregulate the VEGFR2 factor to enhance the formation of new blood vessels. This process promotes the growth of cancer cells, proliferation, and metastasis by providing them with increased nutrients and oxygen. Therefore, the VEGFR2 signaling pathway has a significant association with the development of cancer and its progression [10].

The *in silico* study is a revolutionary strategy in drug discovery with a cost-effective approach of screening therapeutic agents. Molecular docking, molecular dynamics (MD) simulation, and ADMET analysis help assess binding energies and

pharmacokinetics without labor-intensive experiments. In studies of BC, targeting VEGFR2-driven angiogenesis is considered essential, as it ensures tumor growth and metastasis [11].

Some of the traditional therapies employed in the treatment of breast cancer include chemotherapy, surgery, radiotherapy, and hormonal treatment. However, high heterogeneity in BC, resultant side effects, and onset of multidrug resistance typically result in complications leading to an unfavorable prognosis. Therefore, naturally occurring food products and medicinal plant products are gaining more interest as complementary or alternative treatment methods. Natural compounds have the potential to modulate epigenetic events and reverse changes that lead to the onset of cancer. Phytochemicals do this by triggering cell cycle arrest, apoptosis, and reactivation of tumor suppressor genes through the action on certain transcription factors, growth factor pathways, and kinases [12,13].

VEGFR-2 inhibitors are classified as Type I (e.g., sunitinib), which occupy the active ATP site, and Type II (e.g., sorafenib, regorafenib, tivozanib), which act at an allosteric site in the inactive (DFG-out) conformation. Type-II inhibitors are more selective and possess long-lived binding, but can induce hypertension. Quinoxaline and piperazine scaffolds have been reported to possess both VEGFR-2 inhibition and antihypertensive activities. Directed by four predominant pharmacophoric elements of Type-II inhibitors and a strategy for molecular hybridization, piperazinylquinoxaline derivatives were synthesized. In these derivatives, the quinoxaline binds to the ATP site (CYS919/GLU917), piperazine spans over to the DFG domain, an amide moiety acts on GLU885/ASP1046, and a hydrophobic terminal moiety increases receptor affinity [14].

Natural compounds from various plants have contributed to the development of anticancer drugs with fewer adverse side effects [15]. Legundi plants (*Vitex trifolia*) are known to produce a diverse range of diterpenoids with antioxidant, cytotoxic, and trypanocidal activities [16]. Additionally, this plant is rich in flavonoids and has been traditionally used to alleviate rheumatic pain and swelling. Pharmacologically, *Vitex trifolia* also exhibits antibacterial properties and hepato-protective activity [17]. The potential of *Vitex trifolia*, a medicinal herb with anti-inflammatory, antioxidant, and anticancer activity, was discovered by researchers to prevent the target protein. Flavonoids have been known to block tumors due to the presence of anticancer activity within them, and thus they are beneficial for BC treatment [18]. Flavonoids exhibit potent anticancer activity against BC. Their mechanisms of action include triggering apoptosis, autophagy, cell cycle arrest, and inhibition of proliferation and invasion. Particularly, flavonoids modulate cancer-related processes, such as mTOR, which regulates cell growth, metabolism, and survival indicating their clinical utility in BC [19,20]. A study Isolated pure compounds artemetin, vitexicarpin, and pendaletin from *Vitex negundo* exhibited strong antiproliferative activity against HepG2 and MCF-7 cells through ROS induction, apoptosis, and activation of caspases. Molecular docking validated them as anticancer drug candidates [21].

A previous study revealed that *Vitex trifolia* (legundi) acts on breast cancer by causing cytotoxicity in MCF-7 cells primarily through its n-hexane fraction. Active constituents such as flavonoids and terpenoids can induce apoptosis and inhibit cell growth in cancer cells, indicating that it could be a natural anticancer drug [22]. In another study, methanol and petroleum ether leaf extracts of *Vitex trifolia* were screened for cytotoxicity against Vero cells and MCF-7 breast cancer cell lines. Both of them inhibited MCF-7 cell viability in a dose-dependent manner, but petroleum ether extract was highly cytotoxic. Low toxicity was obtained in Vero cells. These results indicate that *Vitex trifolia* can be a promising chemotherapeutic agent in breast cancer management [23].

This research intends to assess the VEGFR2 inhibitory activity of *Vitex trifolia* phytochemicals as an important regulator of angiogenesis in BC, presented in S1 Fig in S1 File. Through computational methods like molecular docking, interaction studies, density functional theory (DFT), and MD simulation, this research looks for potential VEGFR2 inhibitors that can prevent tumor development and metastasis. The choice of *Vitex trifolia* phytochemicals was based on structural similarity to known pharmacophores of VEGFR2 inhibitors, especially in Type-II inhibitors like sorafenib and tivozanib [14]. Major pharmacophoric features like hydrogen bond donors/acceptors interacting with GLU885 and ASP1046, aromatic systems engaging the hinge area (CYS919/GLU917), and hydrophobic groups augmenting receptor occupancy are frequently

found in flavonoids and terpenoids of *Vitex trifolia* [24]. These features indicate the potential of its phytoconstituents to be ATP-competitive or allosteric VEGFR2 inhibitors. Accordingly, this work establishes a mechanistic basis for investigating *Vitex trifolia* as a valuable lead to selective and efficacious VEGFR2-targeting molecules for BC treatment.

## 2. Methodology

### 2.1 Protein retrieval and preparation

The crystal structure of the VEGFR2 (kinase domain) in PDB format (PDB ID: 4AG8) at 1.95Å [25] was retrieved using the Protein Data Bank (PDB) (https://www.rcsb.org/) database maintained by the Research Collaboratory for Structural Bioinformatics (RCSB). The crystallographic structure of VEGFR in PDB format was subjected to the Protein Preparation Wizard panel of Schrödinger Suite 2020−3 in Maestro 12.5 for protein preparation. The protein was processed by adding hydrogens, filling the missing loops and side chains using Prime, and generating heteroatom states using Epik. Subsequently, the H-bond network was optimized, and water molecules were removed. Lastly, the OPLS3e force field was employed to reorient side-chain hydroxyl groups and improve potential steric conflicts. The minimization is confined to the input protein coordinates by a predetermined Root Mean Square Deviation (RMSD) tolerance of 0.30 Å [26].

### 2.2 Drug selection and compounds retrieval

A total of fifteen compounds of *Vitex trifolia* used in this study were retrieved from previous literature and their 2D and 3D structure constructed were using ChemDraw Professional 16.0 and Chem3D, respectively. Axitinib which was also a co-crystal ligand (CCL), embedded in the crystal structure of the protein (VEGFR2) was used as a reference drug for comparison. The compounds were then prepared using the Ligprep panel form Maestro 12.5, employing the OPLS3e force field, at pH $7.0 \pm 2$ with Epik applied to generate heteroatom states [27]. Heteroatom refers to the protonation tautomeric forms of heteroatoms (such as nitrogen, oxygen, and sulfur) under the specified pH range, ensuring accurate representation of their chemical states for docking. The Ligprep interface was configured to include stereoisomer computation using the Ligprep task, retaining specific chirality (varying other chiral centers) and generating at least 10 stereoisomers per ligand [28].

### 2.3 Receptor grid generation

To define the binding cavity of the targeted protein for docking and to remove the embedded CCL from that site, the "Receptor grid Generation" panel in Maestro 12.5 was used. All compounds were then docked into the generated grid structure, representing the binding cavity of the protein.

### 2.4 Docking analysis

All compounds were docked into the binding pocket using Maestro 12.5. The grid file was imported, and the prepared compounds were loaded into the Ligand Docking panel. Docking was performed using flexible docking with XP (Extra Precision) mode, incorporating XP descriptor instructions for higher accuracy. Additional settings involved calculating per-residue binding scores within a 12.0 Å radius and RMSD for input compound geometries. After dock analysis, docked structures were combined and saved in PDB format, while docking scores were extracted and saved in CSV file format to be analyzed.

### 2.5 Visualization of docking

The saved PDB files of dock complexes were visualized in Discovery Studio Client 2021(https://discover.3ds.com/discovery-studio-visualizer-download) to generate 2D structures of top hit compounds that were docked in the binding cavity of the targeted protein. The same PDBs were then imported into PyMOL-2.4.0 to create 3D structures for visualization [29].

## 2.6 Structural Interaction Fingerprinting (SIFt) analysis

Structural Interaction Fingerprinting analysis was used to predict the interaction of compounds with protein residues in the binary interaction format, providing insight into the interaction pattern of complexes. In this study, fifteen compound complexes that docked within binding cavities of VEGFR2, along with CCL, were used to generate the interaction fingerprints. The SIFt analysis was performed using the "Interaction Fingerprints" panel from the Schrödinger suite 2020−3, where the docking output file was imported, and the important interactions (polar residues, hydrophobic residues, hydrogen bond acceptor, hydrogen bond donor, and aromatic residues) were selected. After analysis, the results were exported as a CSV file, where the interactions were represented in the binary form 0 and 1, which depicted the absence and presence of interaction, respectively [30,31].

## 2.7 DFT studies (MESP/HOMO/LUMO analysis)

The DFT calculations were slightly modified using the previously designated procedure [32]. Utilizing the Gaussian 09 Revision D.01 with the default configuration, all calculations in the SVP basis set utilized the B3LYP function. Using this theory, the electronic structure of atoms and molecules can be effectively calculated. The present investigation will ascertain the optimized geometric parameters, molecular electrostatic potential (MEP), frontier molecular orbital (FMO), and global and local reactivity descriptors. The checks were examined utilizing Gauss View 5.0.8 software.

## 2.8 MD simulation

The docking simulations of the experimentally identified CCL and the VT-6 compounds from *Vitex trifolia* were further optimized and stabilized using molecular dynamics (MD) simulations, performed with the AMBER20 software suite utilizing the ff14SB force field [33]. LEaP module was assigned field parameters for the solvent, and also prepared the system with the proper structural configuration. Two Cl⁻ ions were added for charge neutralization. The system was solvated with an SPCBOX explicit water model with a 9.0 Å buffer region, which helps simulate a physiological environment. The complex was saved in PDB format, and corresponding files were generated with the coordinates and parameters. Energy minimizations were done in three steps: first, optimizing solvent molecules and ions; second, refining the protein-ligand complex, particularly the binding pocket residues; then finally, full steric clash removal and structural equilibration after a full-system minimize. Preceding that, the system was gradually heated from 0 to 300K under NVT ensemble conditions and then equilibrated under NPT conditions of 300 K and 1 atmospheric pressure, which ensured the stabilization of the protein-ligand complex [34–36]. The simulation protocol could be found elsewhere [37].

To analyze the complex's dynamic properties, a 200 ns MD simulation was performed. CPPTRAJ was used for post-simulation analysis to determine structural stability through root mean square deviation (RMSD), root mean square fluctuation (RMSF), and radius of gyration (Rg), and make sure that the system was converged. Additionally, solvent accessible surface area (SASA) was separately calculated for the ligand, protein, and complex for both CCL and VT-6 compound. Followed by determining the buried SASA (B-SASA) in order to assess the effective SASA shared between ligands and the receptor. Notably, a lower B-SASA value depicted a smaller buried surface area and a reduced interface between complexes, highlighting lower binding affinity and vice versa. The B-SASA was computed using the following equation:

$$B\,SASA\,(\text{Å}) \;=\; 0.5\,(SASA_{Receptor} \;+\; SASA_{Ligand} \;-\; SASA_{Receptor-ligand\ complex})$$

For estimating the binding affinity, MM-PBSA and MM-GBSA calculations were performed using 1000 snapshots taken from the last two nanoseconds of the MD trajectory. ΔG_bind or the binding free energy, was calculated by estimating the total free energy of the ligand-protein complex and subtracting the free energy of the protein and ligand when in the

unbound state. Additionally, energy decomposition analysis was conducted to determine important residues that aid in ligand binding through van der Waals ($E_{vdW}$), electrostatic ($E_{ele}$), polar ($G_{ele, sol}$), and nonpolar ($G_{nonpol, sol}$) solvation energies [37].

## 3. Results

### 3.1 Docking analysis

For the docking analysis compounds were derived from the plant *Vitex trifolia* against the target protein VEGFR2. Docking analysis of all compounds was performed and the docking score was compared with the CCL (Table 1). Among all the compounds, VT-6 showed the highest docking score of −14.611 kcal/mol, which reflects strong binding activity with the VEGFR2 active site. VT-10 also had a good binding, with a docking score of −13.641 kcal/mol. These compounds showed a higher binding affinity for VEGFR2 compared to the standard drug (CCL), which had a −13.541 kcal/mol docking score. These results revealed that VT-6 and VT-10 could be potential inhibitors of VEGFR2, having activity comparable to or even better than the standard drug against the target. All compounds with their 2D and 3D structure and SMILES notation are presented in the supplementary data (S1 Table in S1 File).

### 3.2 Visualization of docking Poses in 2D and 3D conformation

The original pose of CCL that was downloaded from PDB was compared with the redocked pose of CCL that was generated from Maestro 12.5. Fig 1A depicts different interactions like conventional H- bonds (green), water-mediated hydrogen bonds (blue), π-π interactions (pink), and hydrophobic interactions like π-alkyl and alkyl interactions (light pink and purple). Major residues participating in binding are GLU885, ASP1046, CYS1045, and LEU1035 and have a very important role to play in the stabilization of the ligand.

2D structures of VT-6 and VT-10 poses that were generated from the Discovery Studio in which VEGFR2 residues interacted with the compounds which were displayed in different colors (Fig 1B).

3D Visualization of compounds and target proteins within the active site was performed using PyMOL (Fig 2A). The protein residues were colored yellow, and the compounds were cyan. Axitinib was labeled as CCL and formed hydrogen bonds with some important amino acids, namely ASP1046, GLU917, GLU885, and CYS919. Cynaroside labeled as VT-6 showed the best score by forming hydrogen bonding with key amino acids CYS919, GLU917, ILE1044, VAL914, and

**Table 1. Codes, names, and docking scores of the selected compounds as VEGFR2 inhibitors.**

| Sr. no. | Codes | Compound Names | Docking Score kcal/mol |
|---|---|---|---|
| 1 | VT-6 | Cynaroside | −14.611 |
| 2 | VT-10 | Quercetin 7-*O*-neohesperidoside | −13.641 |
| 3 | VT-4 | Luteolin-7-glucuronide | −11.821 |
| 4 | VT-7 | Isoorientin | −10.934 |
| 5 | VT-9 | Quercetin | −9.618 |
| 6 | VT-8 | 5-hydroxy-3',4',6,7-tetramethoxyflavone | −8.752 |
| 7 | VT-1 | Apigenin | −8.19 |
| 8 | VT-3 | Luteolin | −7.768 |
| 9 | VT-5 | Orientin | −7.114 |
| 10 | VT-15 | 7-desmethyl artemetin | −6.852 |
| 11 | VT-11 | Vitexicarpin (casticin) | −3.935 |
| 12 | VT-14 | Artemetin | −3.832 |
| 13 | VT-2 | Vertexin | −3.125 |
| 14 | CCL | Axitinib | −13.541 |

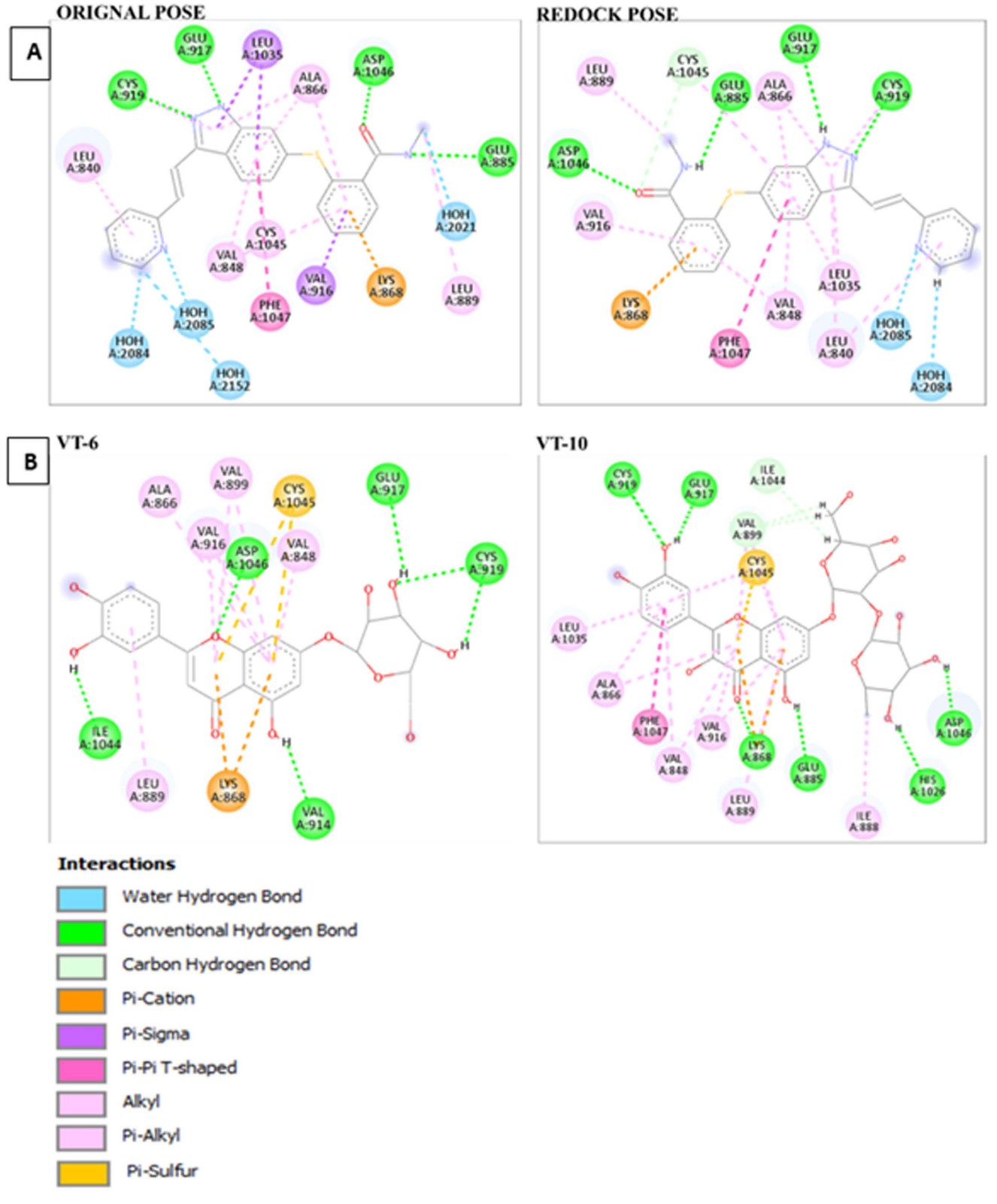

**Fig 1. (A) Comparison of 2D structures of original and re-dock pose of Co-Crystal Ligand. (B)** 2D structures of top hit compounds.

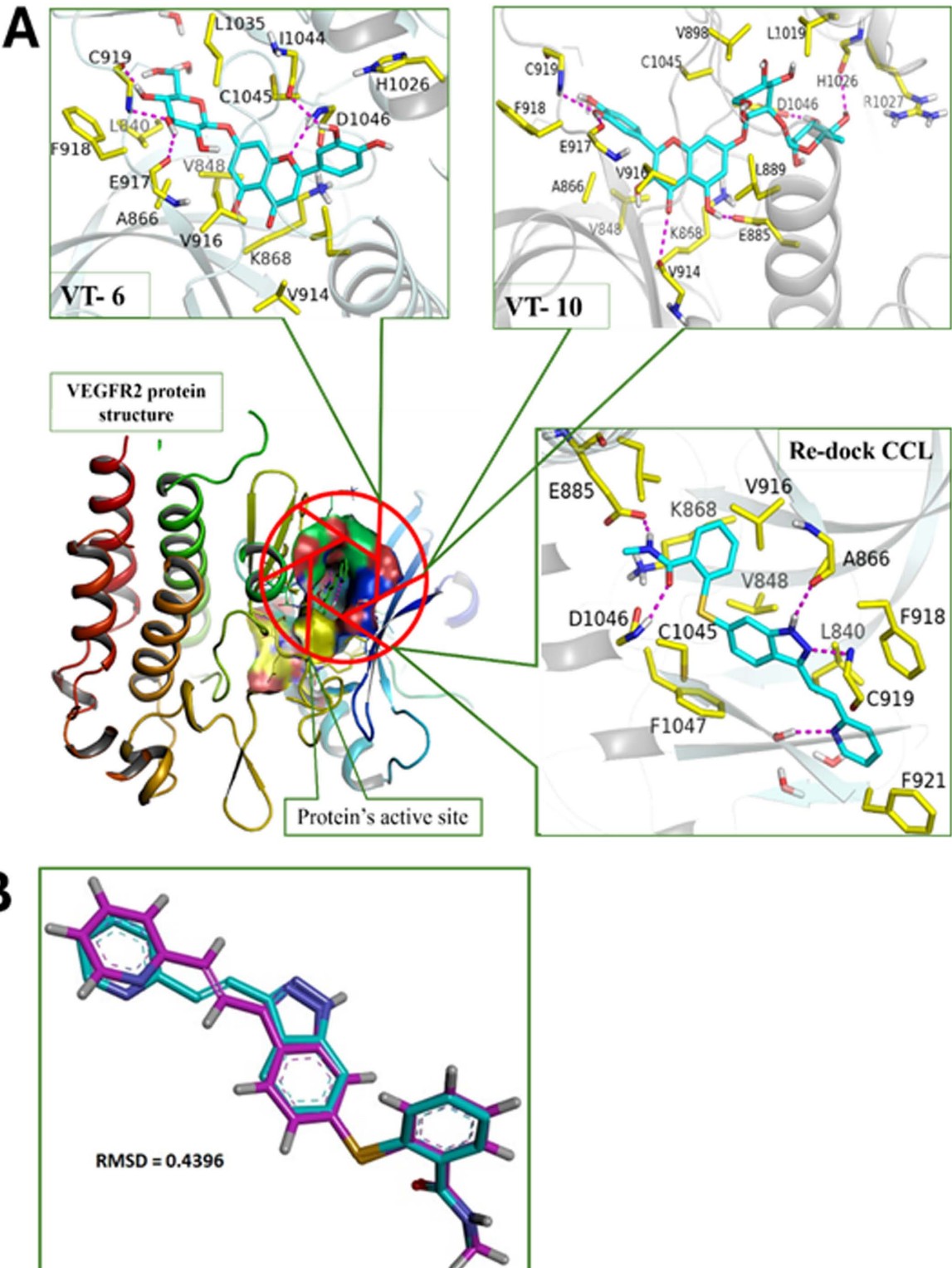

**Fig 2. (A) 3D docking visualization through PyMOL of top compounds in complex with VEGFR2 Protein.** (B) Superimposition of original and redock co-crystal of Axitinib for validation.

ASP1046. Among these amino acids, ILE1044 and VAL914 were more important in inhibiting the target protein. While VT-10 formed hydrogen bonds with HIS1026, ASP1046, GLU885, GLU917, CYS919, and LYS868.The results revealed that both compounds exhibited stronger affinities than CCL, hence, they can be more effective. Hydrogen bond distances and angles describing polar interactions between each compounds and protein residues are presented in Table 2.

With respect to hydrophobic interaction, CCL exhibits contacts with LEU840, VAL848, and ALA866 in the distance ranges of 3.27 Å to 3.93 Å, respectively, which play very important roles for the complex's stability. In contrast, VT-6 also exhibits a superior hydrophobic interaction profile by having crucial residues such as VAL848, ALA866, GLU885, and VAL899 exhibit contact at distance ranges of 3.26 Å to 3.97 Å, signifying stronger and stable hydrophobic contacts. Interestingly, VT-6 performs better than VT-10, which also makes hydrophobic contacts with residues like VAL848, ALA866, and ILE888 but with slightly reduced interaction distances. The overall binding pattern of VT-6, with strong hydrogen as well as hydrophobic bonds, highlights its better docking score, making it the most promising candidate for further study. Hydrophobic interactions showing bond distances between compounds and protein residues are presented in S2 Table in S1 File.

Since 2Å is the threshold RMSD value for assessing molecular docking accuracy, the obtained RMSD of 0.4396Å in the superimposed structures suggested high docking accuracy. This corresponds to excellent agreement between the co-crystal structure's re-dock and the original pose, thus meaning it is good in predicting the binding conformation as shown in (Fig 2B).

**3.2.1 Surface property analysis of VEGFR2 binding cavity.** Various surfaces were used to analyze the binding cavity of protein which demonstrated the binding interaction of compound and target protein residues. Firstly, the aromatic surface of the cavity depicts the stacking region of the cavity that facilitates stabilizing the compound as shown in (Fig 3). The surface of the H-bond indicated that the Hydrogen bond donor and acceptor sites were crucial for the stabilization of compound interactions. The surface of interpolated charge distribution highlighted the strength of interaction and electrostatic impact. The surface of hydrophobicity depicted the interaction of nonpolar compounds that were important to the stable binding cavity. The map of ionizability displayed the sections with maximal acidic and basic areas that could create robust binding interactions with the compound while showing an optimized fit. By using Solvent-Accessible Surface (SAS) can analyze the surface-exposed and buried areas that highlight any regions suitable for binding. This structural representation can help to evaluate the interaction between VEGFR2 and compounds, thus facilitating the identification of an efficient that can be used as a therapeutic target.

## 3.3 Structural Interaction Fingerprinting (SIFt) analysis

Further details on the molecular interactions between the VEGFR2 protein residues and several VT compounds, as well as the CCL compound, are described in the figure presented below (Fig 4). It is important to note that several sites of VEGFR2 protein were totally involved in contact with several residues, namely A848, A868, A885, A1026 and A1046. These residues were used in various interactions including hydrophobic interactions, polar, as well as hydrogen bond interaction which was very crucial in the stabilization of the compound bound receptor.

Comparing the interaction pattern of all the components of VT, it had been found that compounds VT-6 and VT-10 showed the best interaction (Fig 4A). Both compounds engage seven crucial residues A848, A868, A885, A916, A1026, A1046 through different interactions like hydrophobic, polar hydrogen bonding etc. These results suggested that both VT-6 and VT-10 could effectively bound to VEGFR2, with VT-6 exhibited interaction with some of the most significant residues enhancing binding affinity as well. In contrast, CCL compound showed interactions with A848, A868, A885 and A1046. Overall present interactions depicted that these compounds might provide the best suited binding affinity and stability with the VEGFR2 protein, supporting their potential therapeutic applications and further molecular docking.

SIFt profile for the residues of VEGFR2 that interact with VT-compounds, which was consistent with the information in (Fig 4A). In the heatmap (Fig 4B), the horizontal axis showed compounds such as VT-6, and VT-10 while the vertical

**Table 2. Hydrogen bond distances and angles describing polar interactions between each compounds and protein residues.**

| Compound | Residue | Amino Acid | H–A Distance (Å) | D–A Distance (Å) | Donor Angle (°) |
|---|---|---|---|---|---|
| CCL | 919 | CYS | 2.96 | 3.52 | 116.21 |
| CCL | 1046 | ASP | 2.09 | 2.99 | 147.17 |
| VT-1 | 866 | ALA | 2.50 | 3.07 | 118.74 |
| VT-2 | 850 | GLU | 3.35/ 1.73 | 3.70/ 2.64 | 104.36/ 153.19 |
|  | 919 | CYS | 1.74 | 2.68 | 163.98 |
|  | 922 | GLY | 3.51 | 4.03 | 114.37 |
| VT-3 | 868 | LYS | 2.88 | 3.29 | 104.99 |
|  | 914 | VAL | 3.00 | 3.51 | 114.16 |
|  | 1044 | ILE | 2.77 | 3.60 | 144.62 |
| VT-4 | 868 | LYS | 2.64 | 3.14 | 110.85 |
|  | 885 | GLU | 1.67 | 2.58 | 154.95 |
|  | 917 | GLU | 2.27 | 3.22 | 167.94 |
|  | 919 | CYS | 2.49 | 2.98 | 109.13 |
|  | 1046 | ASP | 2.64/ 3.06 | 3.21/ 3.46 | 115.56/ 106.70 |
| VT-5 | 921 | PHE | 1.60 | 2.56 | 165.00 |
| VT-6 | 914 | VAL | 3.09 | 3.76 | 127.83 |
|  | 917 | GLU | 2.29 | 3.11 | 140.79 |
|  | 919 | CYS | 2.66/ 1.84 | 3.58/ 2.80 | 151.37/ 169.60 |
|  | 1044 | ILE | 2.65 | 3.50 | 147.41 |
| VT-7 | 866 | ALA | 2.72 | 3.45 | 133.80 |
|  | 868 | LYS | 2.70 | 3.40 | 126.55 |
|  | 1025 | ILE | 3.07/ 1.81 | 3.98/ 2.75 | 149.90/ 163.85 |
|  | 1026 | HIS | 2.48 | 3.11 | 122.26 |
|  | 1027 | ARG | 1.97 | 2.90 | 150.95 |
| VT-8 | 866 | ALA | 2.04 | 2.91 | 149.99 |
|  | 868 | LYS | 2.57 | 3.01 | 105.90 |
|  | 1046 | ASP | 3.37 | 4.08 | 128.63 |
| VT-9 | 868 | LYS | 3.17 | 3.77 | 118.76 |
|  | 885 | GLU | 1.54 | 2.51 | 166.86 |
| VT-10 | 866 | ALA | 2.62 | 3.19 | 118.26 |
|  | 868 | LYS | 2.95 | 3.55 | 119.27 |
|  | 885 | GLU | 1.73 | 2.62 | 150.15 |
|  | 899 | VAL | 3.26 | 3.63 | 104.55 |
|  | 917 | GLU | 2.40 | 3.34 | 166.04 |
|  | 919 | CYS | 2.71 | 3.23 | 112.21 |
|  | 1026 | HIS | 2.74 | 3.48 | 134.25 |
|  | 1027 | ARG | 3.02 | 3.51 | 111.25 |
|  | 1046 | ASP | 1.81/ 2.84/ 3.39 | 2.78/ 3.29/ 3.94 | 168.81/ 107.22/ 118.49 |
| VT-11 | 919 | CYS | 3.01 | 3.87 | 143.75 |
| VT-14 | 919 | CYS | 3.05 | 3.90 | 143.13 |
|  | 920 | LYS | 3.32 | 3.86 | 115.01 |
| VT-15 | 1025 | ILE | 2.40/ 2.01 | 3.28/ 2.71 | 145.99/ 129.51 |
|  | 1027 | ARG | 2.09/ 2.94 | 3.02/ 3.67 | 152.75/ 129.54 |
|  | 1046 | ASP | 2.89 | 3.74 | 142.10 |
| VT-16 | 885 | GLU | 1.88 | 2.75 | 146.83 |
| VT-17 | 840 | LEU | 2.88 | 3.50 | 124.30 |

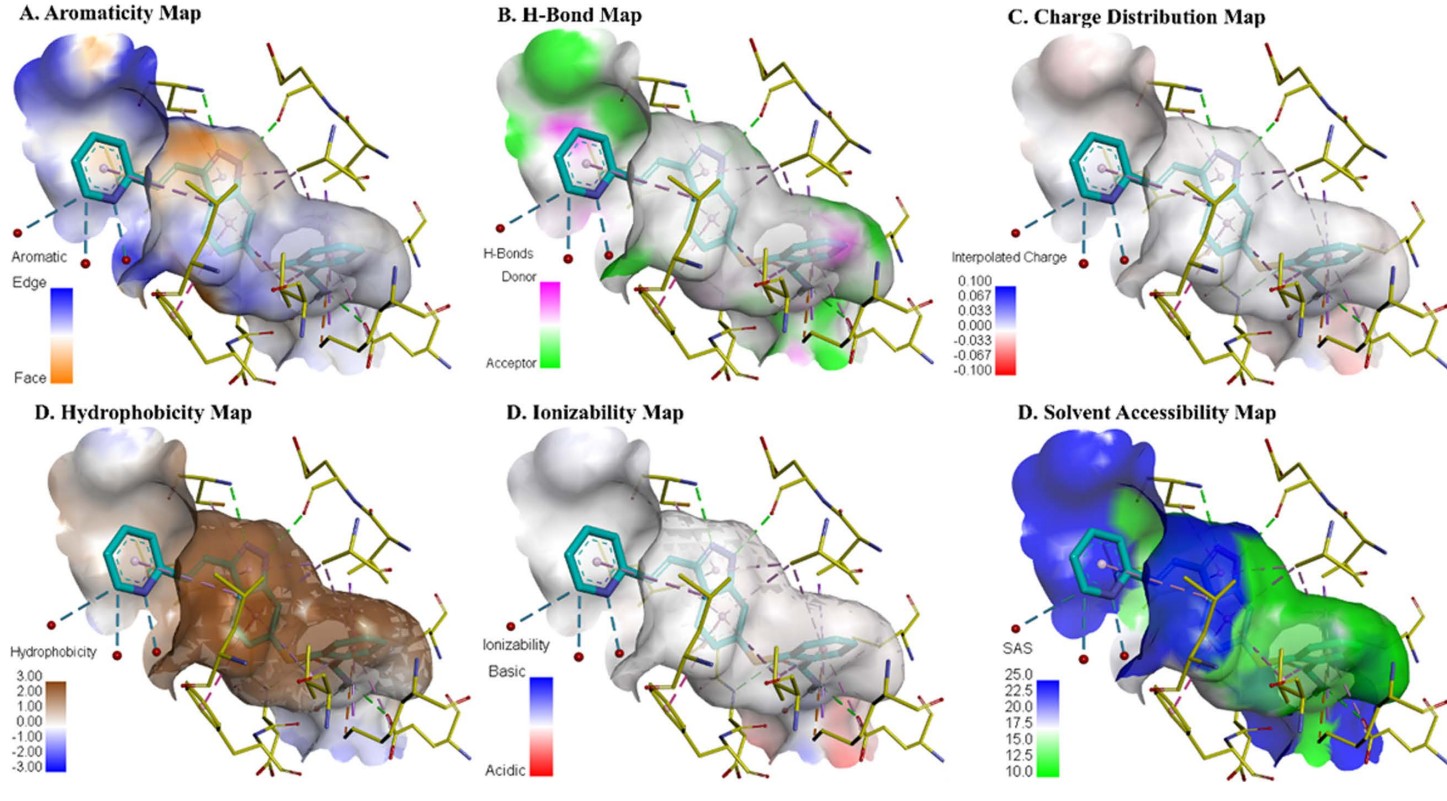

**Fig 3. Surface property analysis of VEGFR2 binding cavity.**

axis showed the residues such as A866, and A889. The heatmap of molecular similarities can also be seen as a distance matrix derived from the SIFt table of VEGFR2 with VT compounds and the CCL compound. The hue in a matrix demonstrates how far apart or how close the compounds can be in terms of indicators to protein residues. The diagonal line represented the highest level of similarity (**Fig 4C**).

### 3.4 DFT analysis

DFT (Density Functional Theory) results have been carried out to determine the nature of the phytochemicals in the gas and water phase indicate the natural physiological state. The electronic distribution was depicted in color-gradient for the MESP (Molecular Electrostatic Potential) surface, which demonstrated that the red color denotes the negative charge which acted as the hydrogen acceptor and gives support to the Van der Waal interaction. Blue color indicated the positive charge, which acted as a strong hydrogen donor. In those figures, the orange and yellow regions indicated the moderate charge, whereas green regions were such neutral regions and less reactive, which supported hydrophobic interaction at the binding sites. Because the spaces between HOMO (Higher Occupied Molecular Orbitals) and LUMO (Lowest Unoccupied Molecular Orbitals) were smaller, the greater energy gap means greater interaction. VT-6 had the same trend in energy gap and was high in bonding score, suggesting that it may be a better VEGFR2 inhibitor. The energy gap of VT-6 and VT-10 was greater than the reference drug CCL, indicating that these inhibitors were the potential therapeutic agents as shown in (**Fig 5**).

DFT calculation was carried out to investigate the electronic properties of VT-6, VT-10, and CCL. Dipole moment values exhibited polarity differences such that CCL had the largest dipole moment (5.5711 Debye). VT-10 contained 5.5137

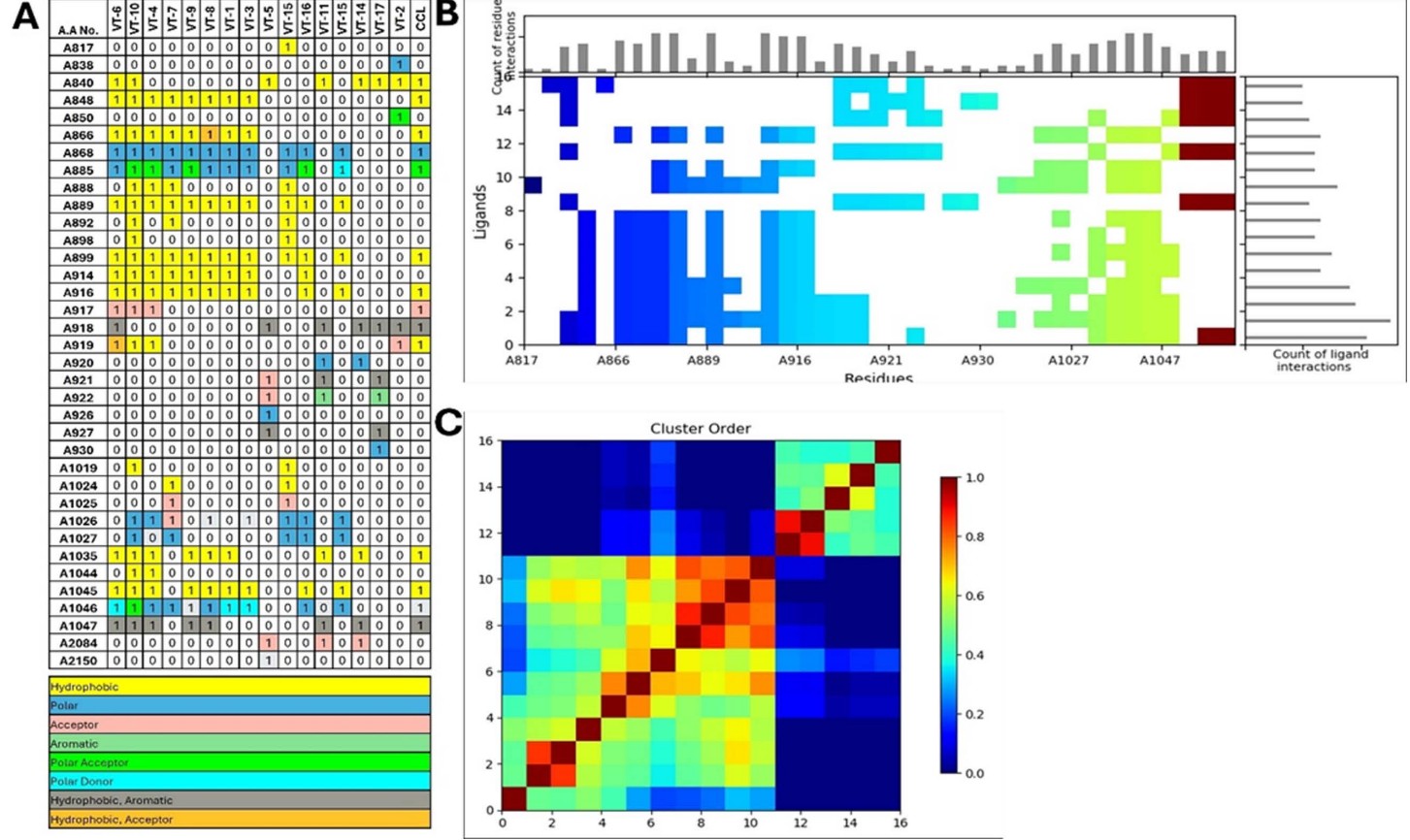

**Fig 4. SIFt Analysis of VEGFR2 with VT compounds and CCL Compound. (A)** Table of SIFt Analysis of VEGFR2 with VT compounds and CCL Compound. **(B)** SIFt Interaction Heatmap of VEGFR2 Residues with VT compounds. **(C)** Distance Matrix Heatmap of VEGFR2 Interactions with VT compounds and CCL Compound.

Debye, which was followed by VT-6 with 5.0823 Debye. HOMO and LUMO energies revealed electron-donating and accepting differences. VT-6 had a HOMO energy of −0.21521 a.u. and LUMO energy of −0.05261 a.u., making up an energy gap (ΔEGap) value of 0.16260 eV, revealing moderate reactivity potential and stability. The comparison revealed a nearly higher value (0.16573 a.u) in the case of VT-10 that signified reduced reactivity and that for CCL as 0.15291 a.u with high reactivity. The ionization potential and electron affinity values also favor the stability of these molecules with CCL having the highest ionization potential (0.22358 a.u.) and electron affinity (0.07067 a.u.). Electronegativity (χ) and electrochemical potential (μ) were compared also, and it was found that VT-6 and VT-10 showed similar electronegativity values, while CCL showed a bit higher value (0.147125 a.u.), which means CCL had a greater electron-withdrawing ability. Besides, the values of hardness (η) and softness (S) indicated that CCL, with the lowest hardness (0.076455 a.u.) and largest softness (13.08 a.u.), was the most reactive. The electrophilicity index (ω) was the largest for CCL (0.14161 a.u.), pointing toward its high electrophilic nature as shown in (**Table 3**). All these observations point toward the electronic nature of the compounds, which are accountable for their biological activity as well as their interactions with target proteins.

These results highlighted VT-6 as a promising candidate based on its excellent electronic properties, well-balanced reactivity, and robust binding interactions. For further pharmacokinetic modification and optimization of its biological

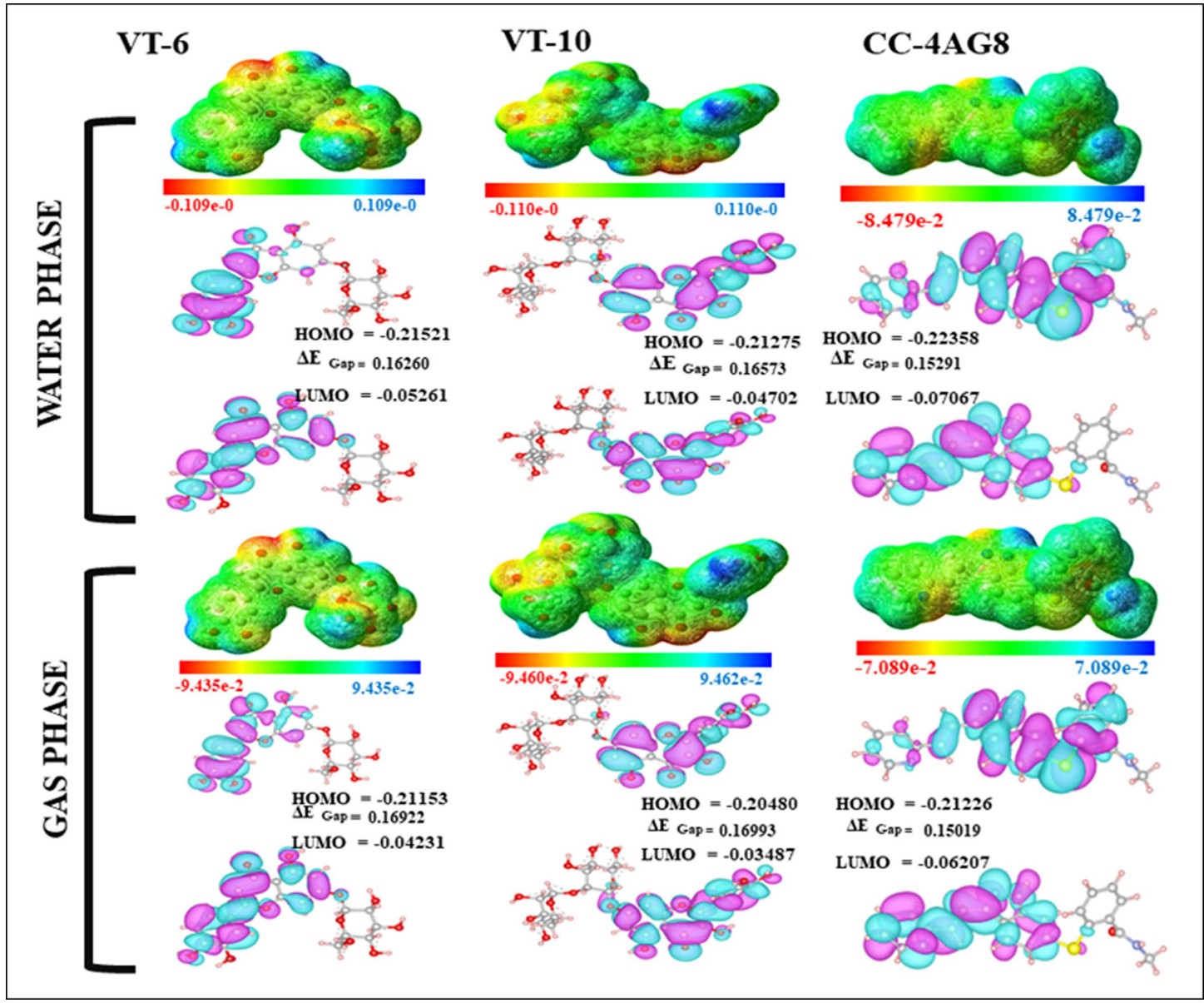

**Fig 5. MESP and HOMO-LUMO analysis of the selected compounds.**

activity, future structural optimization might include the introduction of small polar or halogen substituents to enhance solubility, membrane permeability, or binding affinity. These modifications, based on frontier orbital distribution and electrostatic surface analysis, would facilitate future lead refinement and development.

## 3.5 MD simulation

MD simulations of protein-compound complexes provide for high-resolution temporal behavior, thus offering a context within which the interplay of stability and flexibility at the atomic level and the delineation of interaction strengths can be described. In this research, through 200 ns MD simulations, the binding behavior, stability, and VEGFR-2 inhibitory

**Table 3. DFT analysis parameters of the top hit compounds.**

| Parameters for DFT analysis | Unit | VT-6 | VT-10 | CCL |
|---|---|---|---|---|
| Dipole moment | (Debye) | 5.0823 | 5.5137 | 5.5711 |
| HOMO | (a.u.) | −0.21521 | −0.21275 | −0.22358 |
| LUMO | (a.u.) | −0.05261 | −0.04702 | −0.07067 |
| Energy Gap (ΔEGap) | (a.u) | 0.16260 | 0.16573 | 0.15291 |
| Ionization Potential | (a.u.) | 0.21521 | 0.21275 | 0.22358 |
| Electron affinity | (a.u.) | 0.05261 | 0.04702 | 0.07067 |
| Electronegativity χ | (a.u.) | 0.13391 | 0.129885 | 0.147125 |
| Electrochemical potential μ | (a.u.) | −0.13391 | −0.129885 | −0.147125 |
| Hardness η | (a.u.) | 0.08130 | 0.082865 | 0.076455 |
| Softness S | (a.u.) | 12.30 | 12.07 | 13.08 |
| Electrophilicity ω | (a.u.) | 0.1102 | 0.1018 | 0.14161 |

potential of CCL and VT-6 complexes bound to VEGFR-2 were determined. The results of all key analyses including Root Mean Square Deviation (RMSD), Root Mean Square Fluctuation (RMSF), Solvent Accessible Surface Area (SASA) along with buried SASA (B-SASA), Radius of Gyration (Rg), Principal Component Analysis (PCA), cross-correlation, and binding free energy calculations were compared to investigated which compound may act as the better inhibitor.

The RMSD analysis revealed that both the CCL and VT-6 compounds remain stably bound to VEGFR2 throughout the 200 ns simulation with minimal fluctuations as shown in (**Fig 6**). The protein RMSD of about 2 Å indicated overall structure was stable, and binding pocket RMSD remained low, suggesting that the active site retained its integrity. Additionally, Ligand RMSD showed minimal deviations, confirming that both compounds maintained strong interactions with the protein. The superposition analysis also validated these results since both CCL and VT-6 were found to fit within the binding pocket with minor loop fluctuations. These findings highly indicate that CCL and VT-6 will be stable VEGFR-2 inhibitors, thus being potential candidates for further study.

VT-6 in RMSF analysis showed minimal core helices fluctuation, αC and αD as compared to CCL, indicating that these VT-6 molecules were more stabilized at those sites (**Fig 7**). Significant fluctuation noticed at the flexible regions, such as P-loop and AS/T-loop regions, suggesting high possibilities for conformational changes upon the binding of an inhibitor. VT-6 indicated stability in favor of an inhibitor. The same Rg values for both the compounds seem to point toward global stability, and again, the slightly smaller value of Rg for VT-6 may hint towards a more compact and stable conformation. Hence, in general, it seems that VT-6 had a lower RMSD (RMSF), a more compact binding. Thus, this suggested that VT-6 had a more stable, rigid interaction with VEGFR-2, and can be a promising inhibitor candidate.

Moreover, the SASA analysis revealed that both compounds had stable solvent exposure while the somewhat lower value for VT-6 hints at more compact binding that could be responsible for stronger interactions with the drug target, VEGFR-2 (**Fig 8**). However, to further assess the stability of ligands within the cavity bound at the binding site, the B-SASA calculation was performed, which denoted the parts of SASA that became isolated as VT-6 and CCL interact with the receptor. As mentioned earlier in the methodology section, larger B-SASA suggests greater buried surface area, indicating stronger binding affinity, whereas lower B-SASA suggests fewer interactions and lower binding affinity. The SASA plots presented in **Fig 8** showed that the CCL complex exhibited a B-SASA of 13850.08Å, compared to the VT-6 complex, which exhibited a B-SASA of 13833.30Å. Overall, the SASA analysis highlighted that the VT-6 complex was more dynamic and flexible than CCL. Both complexes exhibited almost comparable binding stability, with CCL possessing higher B-SASA, underscoring its better affinity than VT-6 during 200 ns simulations.

Both MM/PBSA and MM/GBSA analyses revealed VT-6 to as a more potent inhibitor to VEGFR-2, although CCL contributed a higher van der Waals contribution of −82.3 kcal/mol versus −73.6 kcal/mol (**Fig 9**). VT-6 provided more

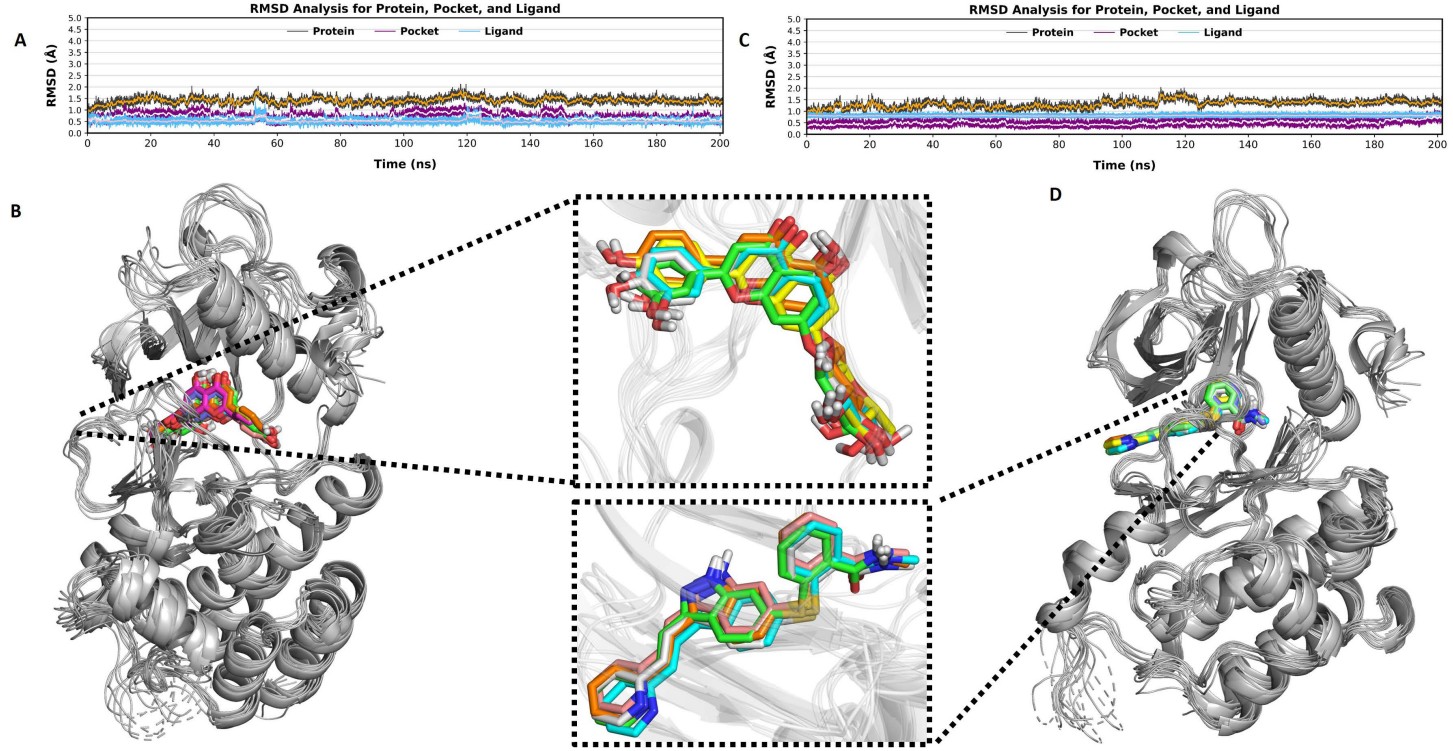

**Fig 6. Analysis of RMSD and Superposition of CCL and VT-6 compounds with VEGFR-2 at the time of Molecular Dynamics Simulation. (A) RMSD Analysis of CCL with VEGFR-2:** The Graph shows Root Mean Square Deviation (RMSD) over a 200 ns simulation, with how the CCL compound (blue line) performs inside the protein binding pocket (purple line) as well as the entire protein structure black line. **(B) Superimposition of VEGFR-2 with CCL:** Dynamics of the interaction between the compound CCL and the binding cavity of the protein during the simulation, including stable and unstable areas. **(C) RMSD Analysis of VT-6 with VEGFR-2**: Changes in RMSD for the VT-6 compound are indicated by a blue line, the binding pocket of the protein by a purple line, and the protein structure by a black line, indicating stability and interaction of the compound with the protein within the 200 ns simulation. **(D) Superimposition of VT-6 with VEGFR-2:** The figure illustrates the dynamic behavior and binding interaction of VT-6 within the protein cavity, underlining major structural alignments and regions of stability.

favorable electrostatic interaction (−53.2 vs. −46.8 kcal/mol) as well as higher free energy of solvation (−60.2 vs. −45.7 kcal/mol), indicating better affinity in the active site (S3 Table in S1 File). The binding free energy of VT-6 came out to be more negative, at −32.5 kcal/mol with MM/PBSA and −34.7 kcal/mol with MM/GBSA, compared to that of CCL, which had binding free energies of −28.1 and −30.6 kcal/mol, respectively. In a nutshell, VT-6 may have the potential as an anticancer drug since its electrostatic and solvation properties were superior as a VEGFR-2 inhibitor.

**3.5.1 Total energy decomposition analysis.** A per-residue energy decomposition analysis was performed to gain deep insight into the molecular interactions of the VT-6 complex. The results are presented in the **Table 4** and **Fig 10**, highlighting that the majority of the residues exhibited more negative total interaction energies. The highest total energy contribution was observed by Glu70 (−3.120 kcal/mol) residue due to stronger electrostatic attraction (−6.7196 kcal/mol). Additionally, Glu102 (−2.869 kcal/mol) and Cys173 (−2.534 kcal/mol) residues also contributed to binding energies with significantly favorable electrostatics and van der Waals forces. A combination of electrostatic and van der Waals interactions was also demonstrated by Phe103 (−2.466 kcal/mol) and Asp174 (−2.362 kcal/mol) residues, further providing strong binding interactions. However, Leu25, Val33, and Val84 residues majorly contributed to hydrophobic interactions, highlighting the significance of non-polar interactions in stabilizing the ligand within the binding site. Moreover, the Lys105 residue also possessed a total energy of −0.041 kcal/mol, underscoring a minimal contribution to direct ligand binding.

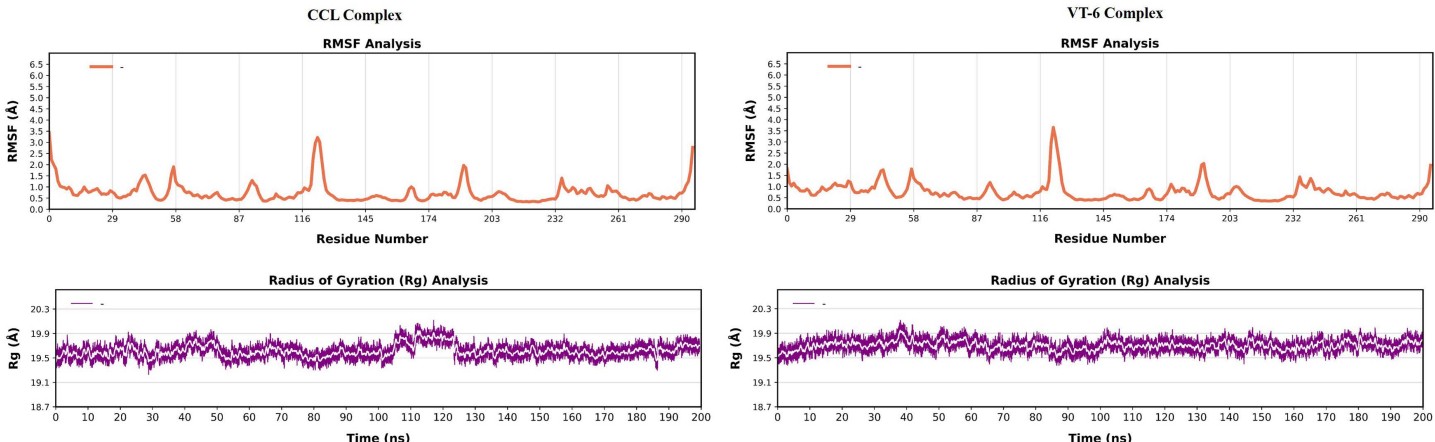

**Fig 7. Comparison between the molecular dynamic simulations of CCL and VT-6: RMSF (Root Mean Square Fluctuation) Analysis of CCL and VT-6 Proteins.** Radius of Gyration (Rg) Over 200 ns Time for CCL and VT-6.

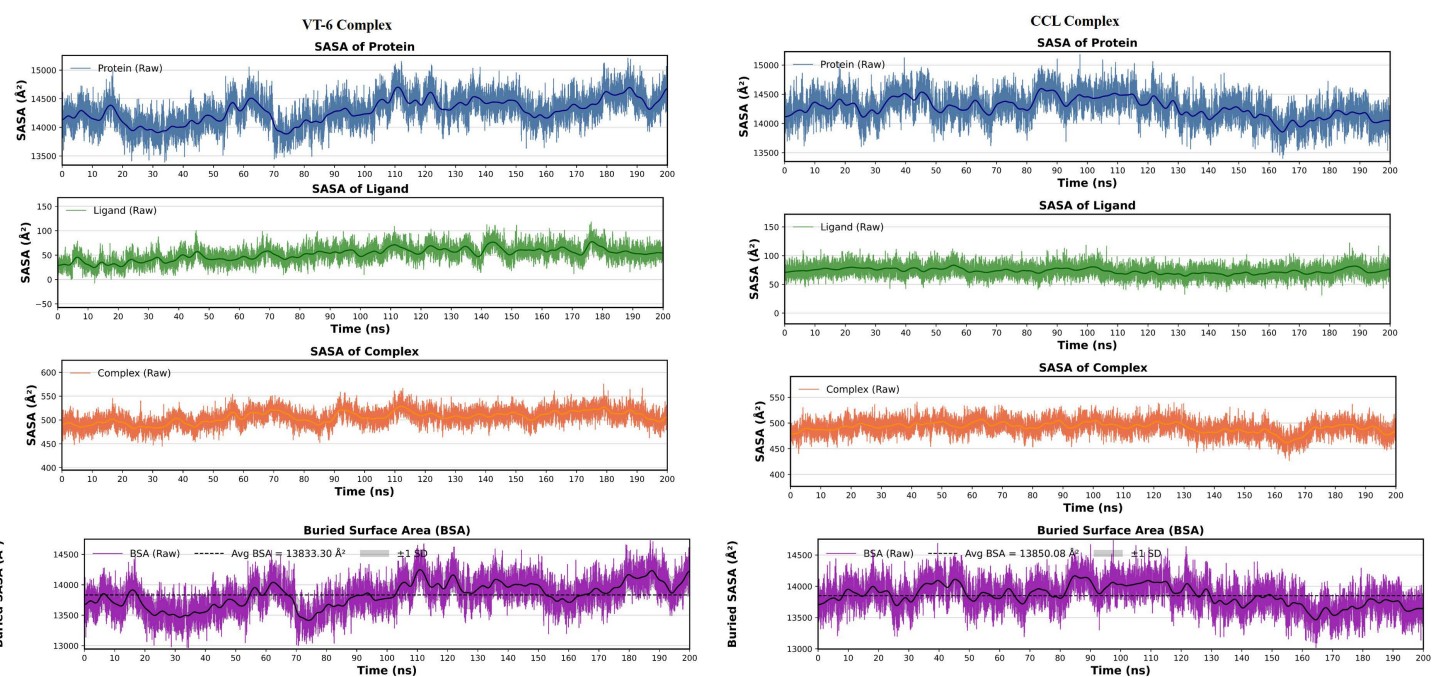

**Fig 8. Comparison between the SASA of CCL and VT-6.** SASA of protein, ligands, and complex for both CCL and VT-6, along with B-SASA calculation for both complexes over a 200-ns trajectory.

Whereas, a few residues also presented positive total interaction energies, such as Val52 (0.005 kcal/mol) and Lys53 (0.099 kcal/mol). These slightly neutral to positive interaction energies were observed due to high polar solvation that outweighed their non-bonded interactions. Overall, the decomposition analysis underscores that the residues of the binding site demonstrated both hydrophobic and electrostatic interactions, significantly playing a crucial role in ligand stability. Notably, residues Glu, Cys, and Phe presented both charged and aromatic side chains, forming favorable interactions with

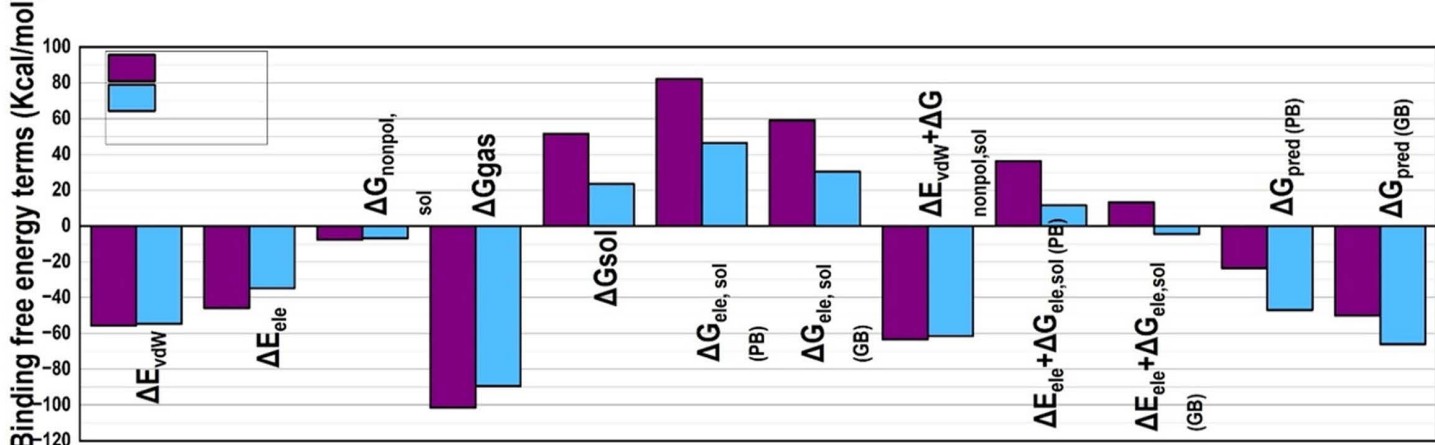

**Fig 9. Comparative binding free energy decomposition analysis for CCL and VT-6 complexes.** The bar chart illustrates various energy components contributing to the binding free energy for CCL (purple) and VT-6 (blue), including- **ΔE_vdW:** Van der Waals energy contributions. **ΔE_ele:** Electrostatic energy contributions. **ΔG_nonpol, sol:** Non-polar solvation energy. **ΔG_gas:** Gas-phase energy. **ΔG_sol:** Total solvation energy. **ΔG_ele, sol (GB):** Electrostatic solvation energy calculated using the Generalized Born (GB) method. **ΔE_vdW+ΔG_nonpol, sol:** Combined van der Waals and non-polar solvation energies. **ΔE_ele+ΔG_ele, sol (PB):** Combined electrostatic energy and Poisson-Boltzmann (PB) solvation energy. **ΔG_pred (PB):** Predicted binding free energy using the Poisson-Boltzmann method. **ΔG_pred (GB):** Predicted binding free energy using the Generalized Born method.

**Table 4. Total energy decomposition analysis of VT-6 complex.**

| Residue | van der Waals | Electrostatic | Polar Solvation | Non-Polar Solvation | TOTAL |
|---|---|---|---|---|---|
| | Avg. | Avg. | Avg. | Avg. | |
| LEU 25 | −1.821 | 0.0259 | 0.1325 | −0.3208212 | −1.98342 |
| VAL 33 | −1.4447 | 0.0806 | −0.1646 | −0.13625064 | −1.66495 |
| ALA 51 | −1.3047 | −0.229 | 0.4209 | −0.0629748 | −1.17577 |
| VAL 52 | −0.5834 | 0.5304 | 0.1059 | −0.00163224 | 0.051268 |
| LYS 53 | −1.8047 | −0.3198 | 2.3304 | −0.10686024 | 0.09904 |
| GLU 70 | −0.4855 | −6.7196 | 4.193 | −0.10831392 | −3.12041 |
| LEU 74 | −0.7807 | 0.2406 | −0.0955 | −0.13770144 | −0.7733 |
| VAL 84 | −1.4 | 0.0497 | −0.2252 | −0.05724216 | −1.63274 |
| VAL 99 | −0.7282 | −0.4231 | 0.1276 | −0.01615032 | −1.03985 |
| ILE 100 | −0.2591 | −0.0753 | −0.079 | 0 | −0.4134 |
| VAL 101 | −1.5958 | 0.0814 | 0.1676 | −0.16228224 | −1.50908 |
| GLU 102 | −0.0693 | −3.3677 | 0.5753 | −0.00808416 | −2.86978 |
| PHE 103 | −1.6913 | −1.4778 | 0.809 | −0.10619352 | −2.46629 |
| CYS 104 | −1.2573 | −1.9487 | 1.5864 | −0.06377688 | −1.68338 |
| LYS 105 | −0.5278 | −0.3505 | 0.9029 | −0.0657072 | −0.04111 |
| GLY 107 | −1.084 | −0.5062 | 0.6234 | −0.19197648 | −1.15878 |
| LEU 163 | −1.5599 | −0.2843 | 0.0553 | −0.15245424 | −1.94135 |
| CYS 173 | −1.373 | −2.7211 | 1.6709 | −0.11103192 | −2.53423 |
| ASP 174 | −1.6816 | 0.8017 | −1.3247 | −0.15789384 | −2.36249 |
| PHE 175 | −1.5143 | −0.2539 | 0.0695 | −0.1036584 | −1.80236 |

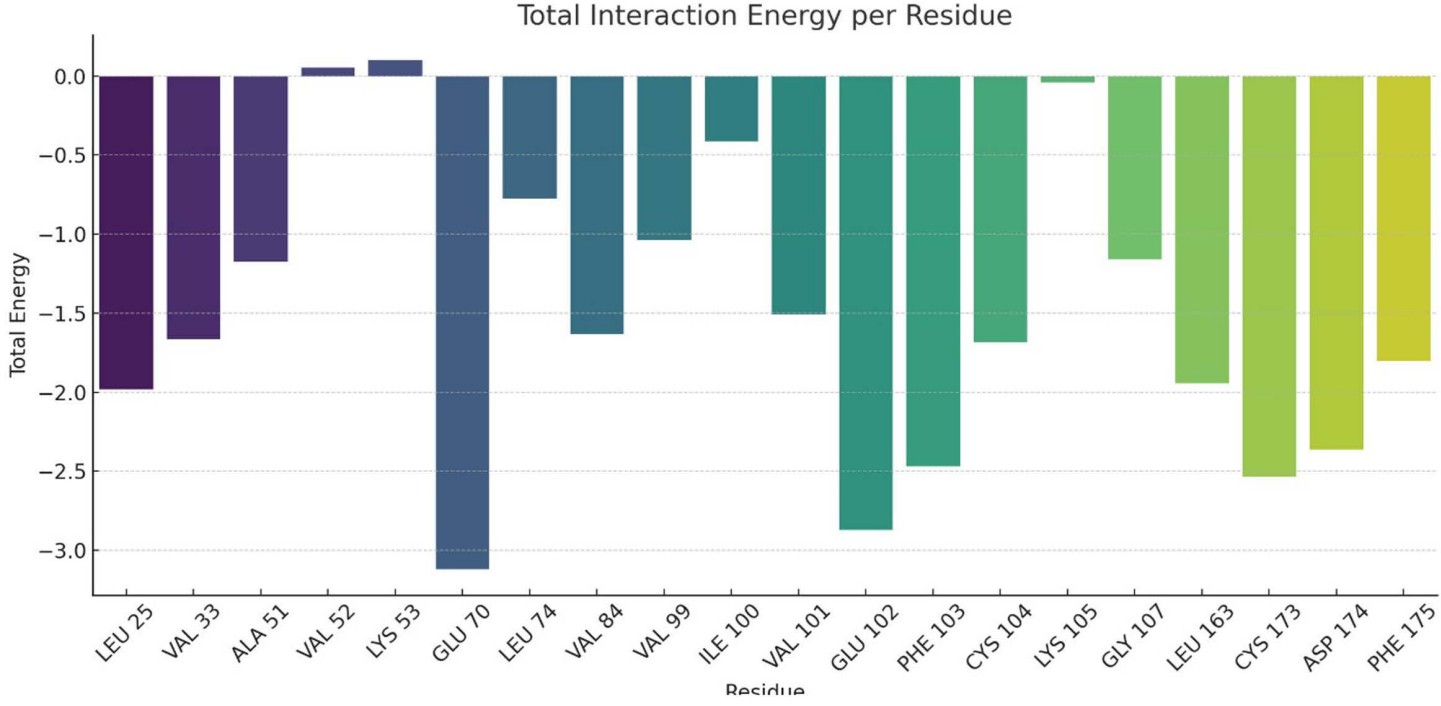

**Fig 10. Total interaction energy contribution by residue in the VT-6 complex.**

the ligand VT-6.0. In addition, per-residue energy decomposition heatmap highlighting van der Waals, electrostatic, polar solvation, and non-polar solvation contributions in protein-ligand Interaction are presented in S2 Fig in S1 File.

Additionally, the conformational stability was further verified by the structural alignment between the apo-protein and holo conformations of proteins was snapped at an interval of 10 ns for each complex. The superimposed structures are shown in S3 and S4 Figs in S1 File. Notably, the RMSD value remained within an acceptable range, thereby supporting that no substantial structural deformities were observed during the 200 ns simulation.

**3.5.2 DCCM, PCA and FEL analysis.** The DCCM (Cross-Correlation Analysis) plot (**Fig 11A**) reveals residue correlations, with red denoting positive correlation and blue negative correlation which exhibit flexibility and stability, respectively. The CCL complex had more correlated movements, indicating flexibility, while VT-6 had less correlated movements, indicating stability. CCL revealed highly correlated residue movements, implying flexibility, while VT-6 revealed lower correlations, implying a more stable binding conformation. PCA plot (**Fig 11B**) displayed conformational changes. CCL had a broader distribution represented flexibility, while VT-6 remained clustered, which demonstrated the stability of the structure. As a result, CCL exhibited more conformation flexibility, whereas VT-6 revealed a denser distribution, signifying a stable conformational state. The FEL plot (**Fig 11C**) illustrated stability based on energy levels. CCL illustrated several energy fluctuations, representing instability and higher energy fluctuations. Whereas VT-6 possessed a lower energy basin (blue), representing VT-6 occupied a well-defined energy minimum, indicating a stable binding mode. **Fig 11D** illustrates the structural deviations. CCL demonstrated more fluctuations (red areas), while VT-6 retains structural integrity with fewer deviations, representing stability. VT-6 had fewer structural variations than CCL, corroborating its stable structure and high binding efficacy. VT-6 showed better stability than CCL, supported by its lesser correlation, efficient PCA clustering, deeper FEL energy minimum, and least structural deviations, making it a potential inhibitor. All these results, along with RMSD, RMSF, SASA, B-SASA and Rg data, suggested that VT-6 imposed structural restriction to stabilize VEGFR-2, implying that VT-6 can be a strong VEGFR-2 inhibitor.

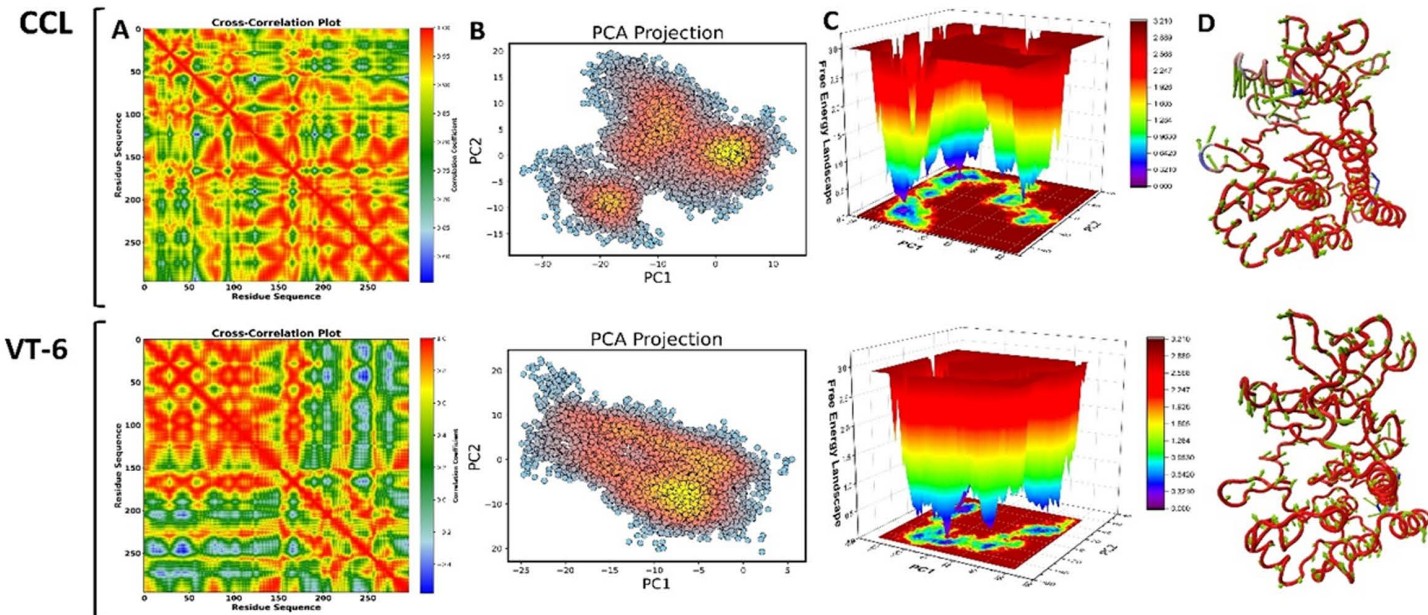

**Fig 11. Comparative analysis of binding dynamics and structural stability of VT-6 and CCL in VEGFR-2 inhibition. (A)** Cross-Correlation Plot of Residue Motion. **(B)** Principal Component Analysis (PCA) Projection. **(C)** Free Energy Landscape (FEL). **(D)** Structural Superimposition of Protein Conformations.

## 4. Discussion

BC poses a significant health issue worldwide due to its uncontrollable cell proliferation and high metastatic potential [38]. VEGFR2 is mainly responsible for regulating angiogenesis, and its participation is vital for tumor growth and progression [39]. It represents a promising therapeutic approach aimed at inhibiting VEGFR2 targeting for BC management. Therefore, an *in silico* approach was used in this study to assess the inhibitory potency of phytochemicals from *Vitex trifolia* against VEGFR2 as potential new antiangiogenic agents.

To investigate the binding affinity and interaction of ligands within the binding cavity, several *in silico*-based studies have employed docking techniques [40–42]. Molecular docking analysis showed that VT-6 (Cynaroside) exhibited a stronger binding affinity than the standard inhibitor (axitinib), with a docking score of 4.611 kcal/mol versus −13.541 kcal/mol respectively. Quercetin 7-O- neohesperidoside (VT-10) also demonstrated a strong binding (−13.641 kcal/mol). In a previous study, molecular docking analysis also confirm the *Vitex trifolia's* potential for targeting inflammation and oxidative stress [43]. Therefore, investigating the binding ability of *Vitex trifolia* phytochemicals to VEGFR-2 might shed light on their wider therapeutic potential in inflammatory disorders, where angiogenesis and VEGFR-2 signaling also have a pivotal role. These findings suggest that phytochemicals from *Vitex trifolia* may effectively target the VEGFR2, exhibiting strong inhibitory potential against this critical tumor-sustaining pathway [44]. In another study, six phytochemicals from *Vitex negundo (L.)* leaves were isloated through GC-MS and docked against WNT signaling proteins associated with colorectal cancer. Compound F demonstrated good binding (–8.7 kcal/mol) having four hydrogen bonds, better than the standard drug Silibinin. These compounds are great candidates for further preclinical investigation [45].

The strong binding affinity of VEGFR2 was further validated by a previous study, which presented nicotinamide-derived compounds as VEGFR2 inhibitors, in which compound 6 displayed high activity with an $IC_{50}$ of 60.83 nM, in addition to strong pro-apoptotic and anti-inflammatory activities. Molecular docking and MD simulations supported its stable binding with VEGFR2, equivalent to the clinically used inhibitor sorafenib [24].

Various studies verify its therapeutic efficacy, demonstrating its antioxidant, anti-inflammatory, hepato-protective, anti-cancer, neuroprotective, antimicrobial, antiviral, antimalarial, antispasmodic, and insecticidal activity [18]. Electronic properties of VT6 and VT10 were derived from DFT calculations. Molecular electrostatic potential (MESP) analysis suggested that such regions of high density of electron were responsible for hydrogen bonding and ligand receptor interaction. This suggested VT-6 would be a good VEGFR2 inhibitor because of its enhanced stability and reactivity, due to the HOMO-LUMO energy gap [46].

MD simulations were further executed to validated the stability of VT-6 in the VEGFR2 binding pocket. The validation of docking studies through MD simulations were also conducted in previously reported literature [47,48]. The structural deviations of the ligand were minimal as shown through RMSD and RMSF analysis and the robustness of ligand protein interactions. Comparative analysis with Axitinib showed lower fluctuation of VT6, restoring its ability as a VEGFR2 inhibitor. Structural interaction fingerprinting (SIFt) analysis indicated that VT-6 interacts with crucial amino acids including ASP1046, GLU917, CYS919, and the others which are essential for the inhibition of the receptor. This is consistent with previous studies highlighting the importance of these residues in VEGFR2 angiogenesis [49]. A novel previous study revealed a series of bis([1,2,4]triazolo)[4,3-a:3',4'-c]quinoxalines exhibited highly effective VEGFR-2 inhibition and anticancer effects. Apoptosis induction of 40.12% was observed for compound 23j and it displayed good ADME characteristics including low BBB permeation, lack of CYP2D6 inhibition, and low plasma protein binding, favoring its application for the treatment of breast cancer. Molecular docking and MD simulation also validated its strong VEGFR-2 binding affinity [50].

Flavonoid-based inhibitors have also exhibited vastly stronger anti-angiogenic properties in comparison with their previous studies. Our finding was consistent with previous reports of the anti-cancer activity of cynaroside, in particular. This also fits in with the current literature on plant VEGFR2 inhibitors, showing that *Vitex trifolia* compounds directly inhibit VEGFR2 [51]. Although promising computational results, this study recognizes the need for experimental validation utilizing *in vitro* and *in vivo* assays. Future work should aim to improve bioavailability and lower the potential to cause cytotoxicity through structural analogs of VT-6. Furthermore, the synergistic effects of combination therapy in the context of VT-6 and current chemotherapeutic agents may be expanded to the treatment of BC.

## 5. Conclusion

This study concludes that the *Vitex trifolia* phytochemicals demonstrated excellent drug potential against the VEGFR2 factor in breast cancer development. Cynaroside (VT-6) and Quercetin 7-O-neohesperidoside (VT-10) demonstrated a higher binding affinity than the control drug. MD simulation data analysis confirmed VT-6 stability with limited fluctuations, while SIFt analysis reported significant residues having stable interactions. Additionally, the DFT analysis highlighted the superior reactivity and stability of VT-6. VT-6 emerged as a promising and effective VEGFR2 inhibitor in the development of breast cancer and reported promising inhibitory activity and therapeutic potential. Nonetheless, since this work relies only on computational approaches, experimental confirmation is required. Further research can look into structural optimization, *in vitro*/*in vivo* screening, and testing against other cancer-related targets or natural sources to expand therapeutic utility. Combination of multi-target profiling and other natural sources could facilitate the identification of more effective and synergistic compounds for targeted BC treatment.

## Supporting information

**S1 File. Supplementary data.**
(DOCX)

## Acknowledgments

We are thankful to Princess Nourah bint Abdulrahman University Researchers Supporting Project number (PNUR-SP2025R420), Princess Nourah bint Abdulrahman University, Riyadh,Saudi Arabia.

## Author contributions

**Conceptualization:** Muhammad Umer Khan, Alaa S. Alhegaili, Nawal Alsubaie.

**Data curation:** Arooj Fatima, Saooda Ibrahim, Qasim Ammar, Raima Rehman.

**Formal analysis:** Arooj Fatima, Saooda Ibrahim, Raima Rehman, Muhammad Ikram Ullah.

**Investigation:** Muhammad Umer Khan, Qasim Ammar, Raima Rehman, Nawal Alsubaie, Muhammad Ikram Ullah.

**Methodology:** Saooda Ibrahim, Iqra Khurram.

**Project administration:** Muhammad Umer Khan, Alaa S. Alhegaili, Nawal Alsubaie.

**Resources:** Muhammad Umer Khan, Alaa S. Alhegaili, Raghad S. Alhejaili.

**Software:** Muhammad Umer Khan, Iqra Khurram, Qasim Ammar, Raima Rehman, Muhammad Ikram Ullah.

**Supervision:** Muhammad Umer Khan, Alaa S. Alhegaili.

**Validation:** Iqra Khurram, Raima Rehman, Raghad S. Alhejaili.

**Visualization:** Saooda Ibrahim, Raghad S. Alhejaili.

**Writing – original draft:** Arooj Fatima, Raghad S. Alhejaili.

**Writing – review & editing:** Muhammad Umer Khan, Nawal Alsubaie, Muhammad Ikram Ullah.

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
