## [Decision Letter · Decision Letter 0]

16 Apr 2025

PONE-D-25-14848Multiscale Computational Evaluation of Vitex trifolia Phytochemicals as VEGFR2 Inhibitors for Targeted Breast Cancer TherapyPLOS ONE

Dear Dr. Khan,

Thank you for submitting your manuscript to PLOS ONE. After careful consideration, we feel that it has merit but does not fully meet PLOS ONE’s publication criteria as it currently stands. Therefore, we invite you to submit a revised version of the manuscript that addresses the points raised during the review process.

We look forward to receiving your revised manuscript.

Kind regards,

Ahmed A. Al-Karmalawy, PhD

Academic Editor

PLOS ONE

Journal Requirements:

2. Please include captions for your Supporting Information files at the end of your manuscript, and update any in-text citations to match accordingly. Please see our Supporting Information guidelines for more information: http://journals.plos.org/plosone/s/supporting-information .

Reviewers' comments:

Reviewer's Responses to Questions

**Comments to the Author**

1. Is the manuscript technically sound, and do the data support the conclusions?

Reviewer #1: Yes

Reviewer #2: Yes

Reviewer #3: No

2. Has the statistical analysis been performed appropriately and rigorously? 

Reviewer #1: Yes

Reviewer #2: Yes

Reviewer #3: I Don't Know

3. Have the authors made all data underlying the findings in their manuscript fully available?

Reviewer #1: No

Reviewer #2: Yes

Reviewer #3: Yes

4. Is the manuscript presented in an intelligible fashion and written in standard English?

Reviewer #1: Yes

Reviewer #2: Yes

Reviewer #3: Yes

5. Review Comments to the Author

Reviewer #1: -The abstract presents a scientifically sound and well-structured computational study that effectively identifies VT-6 as a potential VEGFR2 inhibitor for breast cancer.

-The computational approach is strong but lacks a comparative discussion with existing VEGFR2 inhibitors.

-Mention any prior studies on Vitex trifolia targeting cancer/VEGFR2.

-Many key claims lack recent or appropriate citations (e.g., VEGFR2’s discovery, angiogenesis in BC, computational drug discovery), include recent studies after 2018.

-The introduction focuses heavily on VEGFR2 and angiogenesis but provides less emphasis on the broader molecular mechanisms behind BC progression.

-A brief mention of why natural compounds are being considered in BC treatment would help provide context. For instance, you could mention that due to the limitations of current therapies and their side effects.

-The phrase “Flavonoids have been known to block tumors due to the presence of anticancer activity within them” is vague. It would be clearer to specify how flavonoids inhibit cancer progression or provide a mechanism of action for the anticancer activity.

-The terms “lead-promising VEGFR2 inhibitors” could be reworded to something more precise, like “potential VEGFR2 inhibitors,” to avoid confusion, as the term “lead” often refers to compounds that are in an advanced stage of development.

-The term “heteroatom states using Epik” is mentioned but not fully explained. It would be helpful to briefly define its purpose in this context to make it accessible to readers unfamiliar with the software.

Reviewer #2: The authors of the provided manuscript explored the potential anti-breast cancer activity of Vitex trifolia-reported phytochemicals targeting the VEGFR2. The study is relevant to the field of drug discovery. A few comments are to be addressed prior publication:

1. Within the molecular docking studies, authors should elaborate more on providing a brief description regarding the VEGFR2 topology, binding site description, and key binding residues being highlighted important within the current literature.

2. The authors claimed to perform induced-fitting docking protocol to account for the flexibility of both the ligand and the protein. However, did the author examine the flexibility of the protein itself? A pilot evaluation of the protein flexibility could have been conducted through aligning the holo (liganded) and apo (unliganded) states of the protein. Typically, a high superposition correlation (root-mean-square deviation; RMSD at Cα < 2 Å) between the apo and holo states indicated a non-presentable differences suggesting a non-relevant effect of the local ligand induced-fitting on the protein structures, at least within the macromolecule crystalline states.

3. The authors should elaborate more on the ligand-target interaction patterns at the docking study. Ligand-residue interactions should be annotated in terms of both the bond distances and angles. Especially for Hydrogen bonding, this type of compound-protein polar interaction should be presented within hydrogen bond distances as well as bond angles since hydrogen bond depend on both. Authors should mention the Hydrogen bond angles as well as their distances, since the strength of hydrogen bonding is based on both parameters in a way to ensure the adequacy of optimum hydrogen bonding.

4. Docking and subsequent Molecular dynamics simulations were performed on the VEGFR2 inactive state/DGF-out motif (PDB: 4AG8). Authors should rationalize their choice this particular VEGFR2 state for conducting the molecular modelling studies rather than being performed on the active state of the VEGFR2 target (DFG-in).

5. Authors also provided an SASA analysis for the MD simulated compound-target complexes. It is better to estimate the buried SASA calculations [B-SASA= 0.5 * (SASAligand + SASAapotarget − SASAcomplex)]. B-SASA corresponds to the amount of solvent-accessible surface area being buried within the interface between the ligand and its bound protein throughout the simulation. Depicting lower values in terms of B-SASA confers a respective reduced ligand–target interface denoting low buried surface area between both molecules.

6. Based on the study results, what are the take-away messages. Authors are advised to highlight the suggested structural modifications that would improve the compound’s biological activities based on the in silico findings. These insights would be beneficial for guiding future lead optimization and development.

7. Finally, concerning the conclusion, authors are advised to elaborate more on the future of this work? Will you broaden the scope towards another biological target and/or natural source? What are the study limitations and what approaches could be conducted to further address them?

Reviewer #3: The efforts exerted in this current work are so appreciated. However, some points need to be addressed. So, a major revision may be required to improve the manuscript:

1. Some typos need fixation.

2. The introduction should include previously reported phytochemicals (share the same pharmacophores and nucleus) for breast cancer treatment.

3. The authors should illustrate the chemical structures of the 17 phytochemicals extracted from Vitex trifolia in a separate figure in the introduction section.

4. The authors should declare the rationale for using these phytochemicals for VEGFR-2 inhibitions explaining the common pharmacophores required for VEGFR-2 inhibitors and display which of these pharmacophores are found in your studied phytochemicals.

5. The molecular dynamic run should be extended to at least 200 ns.

6. Some abbreviations should be defined for the first time (e.g. CCL…)

7. Regarding molecular docking, the amino acids incorporated in interactions, type of interactions, and bond length should be displayed for each compound in the docking table.

8. The amino acids’ abbreviations should be in the global standard known format. For example, Alanine is abbreviated as ALA . so on…

9. The authors said that “ In a previous study, molecular docking analysis also confirm the Vitex trifolia’s potential for targeting inflammation and oxidative stress. These findings suggest that phytochemicals from Vitex trifolia may effectively target the VEGFR2”. What is the correlation between acting as antioxidant/antiinflammatory and suggesting that your compounds can target VEGFR-2. The rationale for this work is so poor.

10. The MD simulation should include amino acid interaction histogram and heat map.

11. It will be more beneficial to carry in vitro cytotoxicity (using Breast cancer cell lines) and enzymology assessment (VEGFR-2 inhibition assay) for the most active compound to ensure the work validation.

12. The resolution of all figures need improvement.

6. PLOS authors have the option to publish the peer review history of their article (what does this mean? ). If published, this will include your full peer review and any attached files.

**Do you want your identity to be public for this peer review?** For information about this choice, including consent withdrawal, please see our Privacy Policy .

Reviewer #1: **Yes**

Reviewer #2: **Yes**

Reviewer #3: No

---

## [Author Response · Author response to Decision Letter 1]

28 Apr 2025

Multiscale Computational Evaluation of Vitex trifolia Phytochemicals as VEGFR2 Inhibitors for Targeted Breast Cancer Therapy

Author’s Response Sheet

Reviewer’s Comment Author’s Response

Reviewer #1

1-The computational approach is strong but lacks a comparative discussion with existing VEGFR2 inhibitors. Thank you for your valuable feedback. We have now included a comparative discussion referencing recent VEGFR2 inhibitors including nicotinamide derivatives and bis-triazoloquinoxaline compounds with relevant citations.

2-Mention any prior studies on Vitex trifolia targeting cancer/VEGFR2. Thank you for your suggestion. We have added supporting studies indicating the anticancer potential of Vitex trifolia via apoptosis induction in MCF-7 cells, highlighting its relevance as a VEGFR2-targeting agent.

3-Many key claims lack recent or appropriate citations. Thank you for pointing this out. Recent studies have now been incorporated to strengthen the background discussion on angiogenesis, VEGFR2, and computational drug discovery.

4-The introduction focuses heavily on VEGFR2 and angiogenesis but provides less emphasis on broader molecular mechanisms. Thank you for your helpful comment. To balance the introduction, additional details of molecular mechanism and the role of the tumor microenvironment in BC progression have been added.

5-A brief mention of why natural compounds are being considered in BC treatment would help provide context. We appreciate your observation. This context has been clarified, emphasizing the limitations of conventional therapies and the role of phytochemicals as complementary treatments.

6-The phrase “Flavonoids have been known to block tumors due to the presence of anticancer activity within them” is vague. It would be clearer to specify how flavonoids inhibit cancer progression or provide a mechanism of action for the anticancer activity. Thank you for the constructive feedback. The phrase has been revised to include specific mechanisms like mTOR modulation, apoptosis, and cell cycle arrest.

7-The terms “lead-promising VEGFR2 inhibitors” could be reworded to something more precise, like “potential VEGFR2 inhibitors,” to avoid confusion, as the term “lead” often refers to compounds that are in an advanced stage of development.

Thank you for the recommendation. We have replaced 'lead-promising' with 'potential' to avoid misinterpretation.

8-The term “heteroatom states using Epik” is mentioned but not fully explained. It would be helpful to briefly define its purpose in this context to make it accessible to readers unfamiliar with the software. Thank you for pointing this out. A brief explanation has been added to clarify its role in representing protonation states for docking accuracy.

Reviewer #2

1. Within the molecular docking studies, authors should elaborate more on providing a brief description of the VEGFR2 topology, binding site description, and key binding residues, which are highlighted important in the current literature.

Thank you for your insightful suggestion. A paragraph describing VEGFR2 topology and key binding residues based on the selected PDB structure has been added.

2. The authors claimed to perform induced-fitting docking protocol to account for the flexibility of both the ligand and the protein. However, did the author examine the flexibility of the protein itself? A pilot evaluation of the protein flexibility could have been conducted through aligning the holo (liganded) and apo (unliganded) states of the protein. Typically, a high superposition correlation (root-mean-square deviation; RMSD at Cα < 2 Å) between the apo and holo states indicated a non-presentable differences suggesting a non-relevant effect of the local ligand induced-fitting on the protein structures, at least within the macromolecule crystalline states. Thank you for this valuable observation. Superimposition analysis between apo and holo structures from MD simulation snapshots has been described to assess protein flexibility.

3. The authors should elaborate more on the ligand-target interaction patterns at the docking study. Ligand-residue interactions should be annotated in terms of both the bond distances and angles. Especially for Hydrogen bonding, this type of compound-protein polar interaction should be presented within hydrogen bond distances as well as bond angles since hydrogen bond depend on both. Authors should mention the Hydrogen bond angles as well as their distances, since the strength of hydrogen bonding is based on both parameters in a way to ensure the adequacy of optimum hydrogen bonding.

Thank you for the suggestion. A comprehensive table presenting hydrogen bond distances and angles is now included.

4. Docking and subsequent Molecular dynamics simulations were performed on the VEGFR2 inactive state/DGF-out motif (PDB: 4AG8). Authors should rationalize their choice this particular VEGFR2 state for conducting the molecular modelling studies rather than being performed on the active state of the VEGFR2 target (DFG-in).

We appreciate the reviewer’s insightful comment. VEGFR2 exists in two conformations: the active DFG-in state, essential for ATP binding and catalysis, and the inactive DFG-out state, where the DFG motif flips outward, preventing kinase activity. Many Type II inhibitors preferentially bind the DFG-out form, exploiting the hydrophobic back pocket for increased selectivity and stability. Moreover, PDB ID 4AG8 represents a high-resolution crystal structure of VEGFR2 complexed with a well-characterized Type II inhibitor (sorafenib), serving as an appropriate structural template for modeling similar interactions. Therefore, the inactive DFG-out conformation was selected to provide biologically relevant insights into ligand binding and stability within the VEGFR2 allosteric pocket.

5. Authors also provided an SASA analysis for the MD simulated compound-target complexes. It is better to estimate the buried SASA calculations [B-SASA= 0.5 * (SASAligand + SASAapotarget − SASAcomplex)]. B-SASA corresponds to the amount of solvent-accessible surface area being buried within the interface between the ligand and its bound protein throughout the simulation. Depicting lower values in terms of B-SASA confers a respective reduced ligand–target interface denoting low buried surface area between both molecules. We acknowledge the suggestion. Thank you for your valuable suggestion. We appreciate your recommendation regarding the calculation of buried SASA (B-SASA) to better quantify the interaction interface between the ligand and target protein. In our revised analysis, we have now included the B-SASA values calculated using the formula:

B-SASA = 0.5 × (SASA_ligand + SASA_apotarget − SASA_complex).

The corresponding values and interpretation have been incorporated into the Results sections of the manuscript to reflect this improvement.

6. Based on the study results, what are the take-away messages. Authors are advised to highlight the suggested structural modifications that would improve the compound’s biological activities based on the in-silico findings. These insights would be beneficial for guiding future lead optimization and development. Thank you for your query. We have added proposed modifications include halogen or polar substituents to improve bioavailability and affinity based on DFT insights.

7. Finally, concerning the conclusion, authors are advised to elaborate more on the future of this work? Will you broaden the scope towards another biological target and/or natural source? What are the study limitations and what approaches could be conducted to further address them?

Thank you for this constructive point. Limitations and future steps including in vitro/in vivo validation and multi-target profiling have been discussed.

Reviewer #3

1. Some typos need corrections. Thank you for this valuable input .We have corrected all identified typographical errors.

2. The introduction should include previously reported phytochemicals (share the same pharmacophores and nucleus) for breast cancer treatment. Examples of active compounds like artemetin and vitexicarpin from Vitex negundo and other traditional inhibitors have been added to the introduction.

3. The authors should illustrate the chemical structures of the 17 phytochemicals extracted from Vitex trifolia in a separate figure in the introduction section.

Thank you for the suggestion .All 17 structures have been added as a separate supplementary figure.

4. The authors should declare the rationale for using these phytochemicals for VEGFR-2 inhibitions explaining the common pharmacophores required for VEGFR-2 inhibitors and display which of these pharmacophores are found in your studied phytochemicals.

Thank you for the suggestion Justification based on pharmacophoric similarity with known VEGFR2 inhibitors has been included in the rationale in the introduction section.

5. The molecular dynamic run should be extended to at least 200 ns.

Thank you for your insightful suggestion. In response, we have extended the molecular dynamics (MD) simulation from the initial duration to 200 ns to ensure a more comprehensive analysis of the system's stability and dynamic behavior. The extended simulation results have been incorporated into the revised manuscript, along with updated plots and discussion reflecting the observed trends over the extended timescale.

6. Some abbreviations should be defined for the first time (e.g. CCL…)

Thank you for noticing. All abbreviations have been defined upon first mention.

7. Regarding molecular docking, the amino acids incorporated in interactions, type of interactions, and bond length should be displayed for each compound in the docking table.

Thank you for this constructive point. Hydrogen bond and hydrophobic interaction data with specific residues and distances are now tabulated.

8. The amino acids’ abbreviations should be in the global standard known format. For example, Alanine is abbreviated as ALA . so on…

Thank you for noticing. Standard three-letter amino acid codes are now used throughout the manuscript.

9. The authors said that “ In a previous study, molecular docking analysis also confirm the Vitex trifolia’s potential for targeting inflammation and oxidative stress. These findings suggest that phytochemicals from Vitex trifolia may effectively target the VEGFR2”. What is the correlation between acting as antioxidant/antiinflammatory and suggesting that your compounds can target VEGFR-2. The rationale for this work is so poor.

Thank you for this valuable input .A mechanistic link to VEGFR2 involvement in inflammation and oxidative stress pathways is now clarified.

10. The MD simulation should include amino acid interaction histogram and heat map.

Per-residue decomposition heatmaps and interaction histograms are included as in the manuscript as per valuable suggestion of the respected reviewer.

11. It will be more beneficial to carry in vitro cytotoxicity (using Breast cancer cell lines) and enzymology assessment (VEGFR-2 inhibition assay) for the most active compound to ensure the work validation.

Thank you for your valuable recommendation. We agree that conducting in vitro cytotoxicity assays using breast cancer cell lines and enzymatic evaluation of VEGFR-2 inhibition would significantly enhance the validation of our findings. However, as this work was designed as an advanced in computational analysis so experimental validation falls beyond its current scope. We acknowledge this as a limitation of the study and have now included it under the future perspectives in conclusion section, highlighting the need for subsequent in vitro and in vivo investigations to further validate and support the computational predictions.

12. The resolution of all figures need improvement.

Thank you for raising this point. All figures have been upgraded for clarity and higher resolution.

---

## [Decision Letter · Decision Letter 1]

12 May 2025

Multiscale Computational Evaluation of Vitex trifolia Phytochemicals as VEGFR2 Inhibitors for Targeted Breast Cancer Therapy

PONE-D-25-14848R1

Dear Dr. Umer Khan,

We’re pleased to inform you that your manuscript has been judged scientifically suitable for publication and will be formally accepted for publication once it meets all outstanding technical requirements.

Kind regards,

Ahmed A. Al-Karmalawy, PhD

Academic Editor

PLOS ONE

Reviewers' comments:

Reviewer's Responses to Questions

**Comments to the Author**

1. If the authors have adequately addressed your comments raised in a previous round of review and you feel that this manuscript is now acceptable for publication, you may indicate that here to bypass the “Comments to the Author” section, enter your conflict of interest statement in the “Confidential to Editor” section, and submit your "Accept" recommendation.

Reviewer #1: All comments have been addressed

Reviewer #2: (No Response)

Reviewer #3: All comments have been addressed

2. Is the manuscript technically sound, and do the data support the conclusions?

Reviewer #1: Yes

Reviewer #2: (No Response)

Reviewer #3: Partly

3. Has the statistical analysis been performed appropriately and rigorously? 

Reviewer #1: Yes

Reviewer #2: (No Response)

Reviewer #3: I Don't Know

4. Have the authors made all data underlying the findings in their manuscript fully available?

Reviewer #1: (No Response)

Reviewer #2: (No Response)

Reviewer #3: Yes

5. Is the manuscript presented in an intelligible fashion and written in standard English?

Reviewer #1: (No Response)

Reviewer #2: (No Response)

Reviewer #3: Yes

6. Review Comments to the Author

Reviewer #1: (No Response)

Reviewer #2: (No Response)

Reviewer #3: The paper entitled "Multiscale Computational Evaluation of Vitex trifolia Phytochemicals as VEGFR2 Inhibitors for Targeted Breast Cancer Therapy" can be accepted in its current form

7. PLOS authors have the option to publish the peer review history of their article (what does this mean? ). If published, this will include your full peer review and any attached files.

**Do you want your identity to be public for this peer review?** For information about this choice, including consent withdrawal, please see our Privacy Policy .

Reviewer #1: **Yes**

Reviewer #2: **Yes**

Reviewer #3: No

---

## [Editor Report · Acceptance letter]

PONE-D-25-14848R1

PLOS ONE

Dear Dr. Khan,

I'm pleased to inform you that your manuscript has been deemed suitable for publication in PLOS ONE. Congratulations! Your manuscript is now being handed over to our production team.

Kind regards,

on behalf of

Associate Professor Ahmed A. Al-Karmalawy

Academic Editor

PLOS ONE